

# Characteristics of droughts in Argentina's Core Crop Region

Leandro Carlos Sgroi [1], Miguel Angel Lovino [1,2], Ernesto Hugo Berbery [3], Gabriela Viviana Müller [1,2]

[1]Centro de Estudios de Variabilidad y Cambio Climático (CEVARCAM), Facultad de Ingeniería y Ciencias Hídricas, Universidad Nacional del Litoral, Santa Fe, Argentina

[2]Consejo Nacional de Investigaciones Científicas y Técnicas (CONICET), Santa Fe, Argentina

[3]Earth System Science Interdisciplinary Center (ESSIC)/Cooperative Institute for Satellite Earth System Studies (CISESS), University of Maryland, College Park, MD, USA

*Correspondence to*: Leandro C. Sgroi (lsgroi@unl.edu.ar)

**Abstract.** The current study advances the documentation of dry episodes over Argentina's Core Crop Region, where the production of major crops like wheat, corn, and soybean is most intense and represents the main contribution to the country's Gross Domestic Product. Our analysis focuses on the properties of droughts that include their magnitude, frequency at different time scales, duration, and severity. It is of interest to assess the relationship between those properties and the crop yields. We analyzed 40 years of precipitation and soil moisture at resolutions suitable for regional studies. The analysis of precipitation and soil moisture anomalies is complemented with the corresponding standardized indices estimated at time scales of 3- and 6-months.

Most droughts tend to occur for periods shorter than three months, but a few can extend up to one year and fewer even longer. However, if a multiyear drought experienced breaks, each period would be considered a separate case. Analysis of the frequency distribution indicates that cases of water deficit conditions are more common than instances of water excess. As relevant as the drought duration is its timing and severity. Even short dry spells may have large impacts if they occur at the time of the critical growth period of a given crop. In the core crop region, corn yield is the most sensitive to drought severity. For these reasons, the quantification of severity during the crop-sensitive months is an indicator of what crop yields could be on the next campaign.

## 1 Introduction

Southeastern South America (SESA) is a region where agriculture and cattle ranching are the primary resources and contributors to the region's Gross Domestic Product. In Argentina, for instance, exports of soybean, corn, and wheat, and their derived products accounted for about USD 41.4 billion yearly on average for 2014-2018, (Ministry of Agriculture, Livestock, and Fisheries of Argentina, MAGyP, 2019). Most of the agriculture is rain-fed, with irrigation accounting for less than 3% of the total crop region (Siebert et al., 2013). Thus, crops and stockbreeding are susceptible to climate variability and extremes as they depend highly on natural rainfall. Corn is among the more sensitive crops to water deficits (Minetti et al., 2007), while soybean production requires a middle range of water availability, and tends to be negatively impacted by either wet or dry seasonal extremes (Penalba et al., 2007).



Droughts may have devastating economic and social impacts. Documentation of individual drought events has shown that, indeed, this is the case. The 1988/1989 drought in Argentina was ranked among the worst episodes on record. The cultivated

area was reduced by about 35%, and the crop yield decreased by about 15% resulting in a 44% loss of productivity and, consequently, in high economic losses (IMF, 1990). Another severe drought episode took place during the 2003/2004 austral warm season. The drought started in September 2003 (austral spring), affecting river discharges. By April 2004, the lack of water in the Uruguay River led to the closure of 13 out of 14 turbines at the Salto Grande hydropower plant (La Nación, 2004; Penalba and Vargas, 2008). Yet another severe episode took place during 2008/2009. This drought was at the time one of the

most intense, with reductions of wheat yields of about 50% and leading to the death of 1.5 million cattle heads in Argentina (Barrionuevo, 2009). The drought of late 2011/2012 had substantial impacts on soybean and corn production, causing losses of the order of USD 2.5 billion (Webber, 2012). The more recent drought between November 2017 and April 2018 caused a drop of 33% of soybean production and 15% of maize production during the 2017/2018 season with respect to the previous year (MAGyP, 2018).


Statistical analysis of extreme events in SESA has shown that periods of water excess or deficit occur at different time scales, with an inverse relationship between frequency and duration, i.e., more frequent events tend to be shorter-lived. Hence, many observational studies of drought have centered around two approaches. First, studies that are based on monthly data to examine the evolution of droughts at longer time scales. For instance, Minetti et al. (2007) showed that one-month long droughts account

for about 53% of all cases; two-month droughts are present in 28% of all cases, and droughts of three or more months represent less than 20% of the cases. Second, studies with daily data have shown that even relatively short dry spells can have significant impacts if they occur at times when crops are most sensitive to water availability, as is the case during the growing season. These dry spells occur over smaller regions than those observed in monthly data, therefore with a limited damaging effect (Naumann et al., 2008). Dry spell duration is about six days on average in the Humid Pampas, although they increase in length

towards the west (Penalba and Llano, 2008; Llano and Penalba, 2010; Naumann et al., 2012). Longer dry spells also present and increasing gradient from east to west, up to 60 days in the eastern sector, and about 190 days in semiarid west (Llano and Penalba, 2010).

Dry episodes in SESA have experienced decadal and longer time changes. Changes in frequency of dry and wet spells were

reported as early as in the 19th Century in a visionary study by Ameghino (1884). He even proposed that such changes were due to the introduction of land use practices in colonial times, going back to the 17th Century when water-retaining tall grass was replaced by short grass as agriculture started expanding. More recent studies (e.g., Barrucand et al., 2007; Vargas et al., 2011; Magrin et al., 2014) have reported that the frequency of dry events was larger during the first half of the 20th Century, decaying during the second half when a notorious positive trend in precipitation favored the expansion of agriculture towards

the west onto once semiarid regions. Several studies have shown increases in monthly rainfall and reduction in the number of





dry spells during the 20th Century (Penalba and Vargas, 2008; Naumann et al., 2008; Vargas et al., 2011). Interestingly, recent studies (e.g., Krepper and Zucarelli, 2010; Chen et al., 2010; Lovino et al., 2014) have suggested that the positive trend in monthly precipitation may have slowed down in the first decade of the 21st Century. If confirmed, such change could lead to more frequent droughts impacting future agricultural activities.


The cold phase of ENSO (*La Niña*) is widely recognized as an important forcing for the onset and duration of extreme dry periods in SESA (Labraga et al., 2002; Penalba and Vargas, 2004; Silvestri, 2005; Barrucand et al., 2007; Vargas et al., 2011). Yet, the ENSO cold phase forcing alone does not always lead to intense droughts (Chen et al., 2010; Cavalcanti, 2012). As discussed by Seager et al. (2010) and Mo and Berbery (2011), the ENSO signal on SESA droughts becomes more intense and

with a better-defined spatial shape when the cold ENSO phase is concurrent with a warmer than average North Tropical Atlantic. In addition to the remote forcings, regional and local factors may contribute to extreme event modulation once they are initiated (Mo and Schemm, 2008; Müller et al., 2014). The moisture transports and soil moisture conditions are all known to influence the duration and intensity of the events. Not least, persistent atmospheric circulations, like those associated with blocking episodes, may hinder the development of precipitation systems during long periods. A documented case was the 1962

drought when a persistent and intense blocking anticyclone prevented the supply of warm and moist air from Brazil and the Atlantic Ocean leading to drought conditions over most of Argentina (Malaka and Nuñez, 1980).

This study advances the documentation of dry episode properties in SESA for three different crops: wheat, corn, and soybean. Each type has its phenology, and therefore the critical periods when drought may have the most significant impact on

production will be different for each. The documentation is based on precipitation, soil moisture, and their derived standardized indices. Several drought indices have been defined to characterize droughts. The World Meteorological Organization (WMO) recommends selecting a particular index depending on the data available and ease of application (Byakatonda, 2018). It also recognizes the advantages of the Standard Precipitation Index (SPI) for the study of meteorological droughts (Hayes et al., 2011). In addition to the SPI or any precipitation index, other environmental variables may need to be included depending on

the characteristics of the region of study and the climate (e.g., Byakatonda, 2018). Soil moisture is particularly useful in agricultural areas, as they reflect the water content in the upper part of the soil where crops grow.

In this research, we will (a) identify and examine drought events in recent decades, (b) investigate the frequency of events at different time scales, as well as episodes occurring during critical crop growth periods, and (c) relate the severity of droughts

to crop yields. Section 2 presents the region of interest and describes the data and methods. The results and productivity indices are shown in Section 3. Discussion and conclusions are presented in Section 4.



## 2 Methods

### 2.1 Region of interest

Our analysis focuses on SESA (Fig. 1a) and more specifically in the region known as the Core Crop Region bounded by 36-29°S and 65-59°W (red box in Figs. 1b-d), where most (about 80%) of the Argentine production of wheat, corn, and soybean are found. Dark green color points out the regions where the production of each crop is more intense, values of production for each crop are also in Figs. 1b-d. This region includes almost entirely the Provinces of Córdoba and Santa Fe, and part of the Provinces of Entre Ríos, Buenos Aires, La Pampa, Santiago del Estero, and Corrientes (Provinces are identified in Fig. 1a). [All references to seasons correspond to the austral hemisphere.]


Wheat, corn, and soybean have different life cycles that last about seven to nine months (lower bars in Figs. 1b-d). Wheat is planted during late austral fall or early winter (May-Jun) and harvested during summer, and it is most sensitive to water availability during its growth period in spring (Oct-Nov). Planting of corn and soybean occurs in austral spring (Oct-Dec), and both are harvested in the fall. Their most sensitive period takes place during the summer, specifically Dec-Jan for corn and

Jan-Feb for soybeans. Therefore, a year's crop production could be largely impacted even if a dry period lasting just one month or even less occurs during the critical growth period. While these are the crops' traditional cycles, it has become possible to have double-cropping at specific locations, i.e., have two crops with different cycles in one year by making the second cycle shorter. Crop rotation -which also has the advantage of reducing the need for fertilizers- introduces planting of corn or soybean right after the wheat harvest—the second crop results in smaller but still profitable production (Senigagliese, 2004).

### 2.2 Data sets and drought indices

This analysis of droughts focuses on the properties of rainfall (P), soil moisture (SM) and their derived standard indices; that is, SPI for precipitation and SSI for soil moisture (McKee et al., 1993; 1995; Hao and AghaKouchak, 2014; Hao et al., 2014). SPI represents a standardized precipitation anomaly and stands among the most used indices for quantifying and monitoring droughts (Keyantash and Dracup, 2002; Mishra et al., 2009; Hayes et al., 2011).


The monthly precipitation data from January 1979 to December 2018 employed here was developed by NCEP's Climate Prediction Center (CPC), and consists of in situ observations spatially interpolated to a regular $0.5° \times 0.5°$ latitude-longitude grid cell (Chen et al., 2008). Monthly values of soil moisture at a spatial resolution of $0.25° \times 0.25°$ over the same period were obtained from the Global Land Data Assimilation System (GLDAS; Rodell et al., 2004; Meng et al., 2012; Beaudoing and

Rodell, 2019; 2020). Soil moisture is not a direct measure, but it is produced by the Noah land surface model forced by observations. The Noah Model considers four soil layers (0-10 cm, 10-40 cm, 40-100 cm, and 100-200 cm) totaling 2 meters depth for which the SM total column value was determined (Rodell et al., 2004).





An analysis of the relation between drought occurrence and annual crop yields is performed for the provinces of Santa Fe and
Cordoba. Yield data from 1968/69 to 2018/2019 campaigns were provided by MAGyP and are available at
http://datosestimaciones.magyp.gob.ar/reportes.php?reporte=Estimaciones. Crop yield data were detrended in an attempt to
remove the increasing yields resulting from technological and genetic improvements. The detrended series can be better related
to drought characteristics.

Series of P and SM were turned into anomalies by removing their mean annual cycle. SPI and SSI were computed following
the approach of Hao and AghaKouchak (2014) and Farahmand and AghaKouchak (2015), which allows obtaining
nonparametric standardized indices for many climate variables. The standardized nature of both SPI and SSI allows consistent
comparisons among different locations (Keyantash and Dracup, 2002). Here, SPI and SSI are defined for two different time
scales, three and six months, to facilitate monitoring meteorological, agricultural and hydrological droughts. SPI3 (values for
SPI at 3-months scale) reflect wet or dry conditions for short and medium time ranges and provide estimates of the climate
conditions at critical stages of the crops' growth. SPI6 provides information between seasons and can be a reference point of
the start of the anomalous behavior of flows and reservoir levels, which usually have larger time scales than precipitation itself.
Thus, SPI6 is useful to represent hydrological droughts (Lloyd-Hughes and Saunders, 2002). The less variable SSI index, as
defined by soil moisture content, can identify and monitor more directly seasonal agricultural droughts (Hao et al., 2014).
While SPI is widely used for drought monitoring and prediction, SSI produces a reliable representation of drought persistence
(Farahmand and AghaKouchak, 2015).

The SPI and SSI were calculated following a non-exceedance empirical probability function for extreme events (Gringorten, 1963).

$$p(xi) = \frac{i - 0.44}{n + 0.12},\qquad(1)$$

Equation 1 represents the associated probability of non-exceedance for the *ith* element of the series, where *x* is either P or SM,
*i* is the rank of non-zero values of the sample, and *n* is the size of the sample. This probability is then transformed into
Standardized Indices (*SI*) applying the inverse of the standard normal distribution function ($\emptyset$) to the results of p(xi)
(Farahmand and AghaKouchak, 2015) as follows:

$$SI = \emptyset^{-1}(p(xi)),\qquad(2)$$

This approach is applied both to precipitation and soil moisture to create the corresponding indices, SPI, and SSI.

## 2.3 Approach

Water supply is controlled by the physical system, and water demand responds to the needs of the biological system (Redmond,
2002). When the amount of available water cannot meet the demand, a drought event may take place. Droughts are usually
defined as meteorological when there is a precipitation deficit over a period of time. A continued precipitation deficit leads to





agricultural drought when soil moisture does not meet the plants' demand, and then to hydrological drought when runoff is below a threshold value (see, e.g., Dracup et al., 1980). This study focuses on meteorological and agricultural droughts and their impacts on crop yields within the region of interest.

Thresholds $X$ were defined by deviations from the mean values measured in standard deviation dimensions, $X = \overline{x} - \sigma$, where $\overline{x}$ is the mean value for the time series of the variable, and $\sigma$ is the corresponding standard deviation for each point in the grid. As both SPI and SSI are standardized indices, $\overline{x} = 0$ and $\sigma = 1$ in every grid cell of the domain. Thus, the thresholds used in this study for SPI and SSI are a less restrictive $X_1 = -0.5$, which represents half standard deviation and is considered when evaluating drought duration; and $X_2 = -1.0$ that corresponds with one standard deviation of each index. Drought events

detected below $X_1$ ranges from mild to extreme droughts, and the ones below $X_2$ cover moderate to extreme dry episodes (McKee et al., 1995). We also studied the events detected below one standard deviation using anomalies of precipitation and soil moisture. As both are not standardized variables, the thresholds for each grid cell may vary from point to point within the domain, and they are different than the ones used for the indices.

Dry events were analyzed by studying their frequency, duration, severity, and areal extent. Drought frequency (F) indicates the number of droughts during the time of analysis with respect to the total possible cases, in scales of months, seasons, or the critical months periods for crop growth. Therefore, the frequency analysis is performed for the whole period at monthly steps, at seasonal scale accounting for every month of each season; and also, at monthly time steps, particularly for each crop's critical periods. The frequency distribution of drought events also depends on the duration (D), that is, the length in time an

index remains below the threshold. The drought magnitude is defined as the average deficit of an index during the duration of the event. The drought severity (S) is equivalent to the accumulated water deficit on the event (Dracup et al., 1980) and it is defined as the magnitude times the duration, i.e., $S = D \times M$ (see Yevjevich, 1967; Keyantash and Dracup, 2002). The properties of frequency, duration, and severity of droughts are unique to the thresholds that define them. The analysis is completed with the examination of the droughts' areal extent (A).


Low-frequency variability modes in the drought indices were identified using a Singular Spectrum Analysis (SSA) approach (Ghil et al., 2002; Wilks, 2006). SSA decomposes the time series in temporal-empirical orthogonal functions (T-EOFS) and temporal-principal components (T-PCS) and facilitates the interpretation of processes related to interannual modes of climate variability and the cases of drought. First, a low-pass Lanczos filter (Duchon, 1979) with an 18-month cut-off period was

applied to the monthly-mean time series of precipitation anomalies to remove its annual cycle and emphasize its interannual variability. Then, the SSA was used to identify the interannual nonlinear trends and quasi-oscillatory modes. Following Von Storch and Navarra (1995), we choose a window length (W) of 120 months as it does not exceed one-third of the length of the whole period and resolves quasi-periods in the interannual band 1 year < T < 10 years.



# 3 Results

## 3.1 Droughts in the core crop region

### 3.1.1 Spatial analysis

Figure 2 presents the spatial distribution of drought's probability of occurrence for northern Argentina as characterized by precipitation and soil moisture anomalies lower than one time their corresponding standard deviations, as well as by SPI3 and SPI6. The probability of occurrence of drought in northeastern Argentina ranges between 12 and 16% for precipitation (Fig. 2a) and between 16 and 18% for soil moisture (Fig. 2b). In general, the probability of occurrence estimated from precipitation anomalies exhibits slight increases towards the eastern/northeastern sector of the Core Crop Region. In contrast, the pattern for the probability of occurrence of soil moisture anomalies shows a second area of higher values towards semi-arid regions west of the Core Crop Region (see the red rectangle in each panel). It is speculated that soil moisture in many of these areas may be affected by soil degradation and desertification (Révora, 2011; Drovetto, 2018). The occurrence of droughts characterized by SPI3/6 (Figs. 2c and 2d) reveals a homogeneous spatial distribution that is higher than for precipitation, probably reflecting the emphasis on the lower time scales. Inside the Core Crop Region, droughts have probabilities between 16 and 18% for both SPI (see Figs. 2c and 2d). Fig. 2d indicates that hydrological droughts, as represented by SPI6, occur with a probability of 18% towards the north and southwest of the region.

The probability of occurrence of droughts during the crops' critical growing periods provides useful information for decision making. Crops have a stage during growth when they become more sensitive to water availability, and this changes with the type of crop. For instance, the crucial period for wheat occurs in late spring (October and November), for corn, it is during the summer (December and January) and even later for soybean (January and February). Therefore, spring and summer represent the most important seasons in terms of the crops' critical months. Figure 3 presents the probability of droughts characterized by SPI during the corresponding critical growing period of each crop. Each crop type presents many areas where drought frequencies are 18% or higher. Given the shift in critical periods from wheat to soybean, Figs. 3a-c can also be interpreted as the temporal evolution of drought frequency. This result suggests that droughts are more frequent during summer months than during spring, affecting the corn and soybean critical periods more than the wheat critical period.

Figures 3d-f show that longer droughts, as represented by SPI6, have a probability of 18% in particular over all the Core Crop Region. In contrast, shorter droughts characterized by SPI3 tend to be more common towards the west of the region, affecting corn and soybean crops particularly.

### 3.1.2 Temporal variability

Figure 4 presents the time series of precipitation and soil moisture anomalies, as well as the SPI3, and SPI6 indices, area-averaged over the Core Crop Region. As expected, precipitation (Fig. 4a) exhibits higher frequency variability, with intense



peaks on individual months. The interannual variability of rainfall is relatively regular, although a change towards shorter scales is noted around 2000. The non-linear trend has low magnitude, although it reveals a small increase in wet events since 2010. The results should not be interpreted as if there was a small interannual or lower frequency variability but rather that other variables will be able to depict such variabilities better. This is the case of the SPI indices and soil moisture (Figs 4b-d),

which better identify wet and dry periods and their interannual variability. Soil moisture acts as a physical filter in the sense that it filters the highly variable precipitation into a lower frequency signal. Notably, as the SPI time scale increases (from 3 months to 6), the variability of the series is reduced (see Figs. 4b and 4c).

Table 1 summarizes the dominant modes of interannual variability for precipitation, soil moisture, SPI3, and SPI6. Interannual

variability modes are found in two distinct bands: one with decadal periodicities and the other close to 3 years. Decadal cycles in SPI and soil moisture series are closely related and contribute to the formation of the dry periods 1987-1991, 1994-1999, and 2004-2013 (see Figs. 4b-d). Trends appear in precipitation anomalies and SPIs, explaining different percentages of the total variability of the series. Short-term cycles of interannual variability contributed to the formation of drought episodes and extreme wet events in all the study variables (see Figs. 4a-d). Interannual modes can explain more than 55% of the total

variability of the 18-month filtered precipitation series and 37% of the soil moisture variability. All time series in Figure 4 present a 2.3-year cycle, which shows higher amplitudes around 2000 and contributed to the formation of wet events in 2001, 2003, 2005, and 2007, as well as the droughts in 2008-2009, 2011-2012, and 2017-2018. This result agrees with Lovino et al. (2018a, b), who suggested that short-term variability (2.5- to 4-year periods) in precipitation exhibits a large increase in amplitude after 2000, even during moderate ENSO phases. They speculated that this difference may be explained by an

increase in heavy rainfall resulting from greater atmospheric instability and water vapor content (see Re and Barros, 2009; Penalba and Robledo, 2010).

### 3.1.3 Frequency distribution

Histograms of the different indices (precipitation anomalies, soil moisture anomalies, SPI3, SPI6, SSI3, and SSI6) were prepared to analyze the distribution of wet and dry periods over the Core Crop Region. Fig. 5 reveals asymmetries for most of

the distributions. As we are dealing here with anomalies and standardized indices, a right-skewed histogram indicates more cases of water deficit conditions than water excess conditions, while a left-skewed histogram indicates the opposite. Five out of six histograms present a right-skewed distribution indicating that drought episodes are more common than wet events over the region.

The precipitation and soil moisture anomalies display right-skewed histograms (Fig. 5a) with different kurtosis, or propensity to produce outliers (Westfall, 2014). The precipitation histogram (blue hatched) exhibits extreme events, which are related to a higher kurtosis (see inset in Fig. 5a) and heavy-tailed distribution (Westfall, 2014). The soil moisture histogram shows a more compact distribution with low kurtosis and light-tailed histograms. This indicates that weak water deficit events are more





frequent (e.g., about 150 events are found in the range -10 to 0 mm). On the other hand, a wider departure from the mean for

precipitation histogram indicates that extreme dry events may occur although their frequency is low, revealing, on the one
hand, the need to use multiple indices and, on the other, the complexity of their simultaneous interpretation.

Figs. 5b and 5c show histograms for SPI and SSI indices at time scales of 3 and 6 months. Histograms present similarity
between each distribution, and kurtosis reveals no preference in the tails that may be associated with the standard nature of the

indices, as can be seen from kurtosis values in the insets. SPI3 and SSI3 (Fig. 5b), as well as SSI6 (Fig. 5c), present a right-
skewed distribution, indicating a higher frequency of water deficit events. It is only in the longer time scales (SPI6; blue
hatched histogram) where a left-skewed distribution is found.

In an attempt to have a better understanding of the seasonal distribution of dry events inside the Core Crop Region, seasonal

boxplots were built for precipitation and soil moisture anomalies (Fig. 6). The use of anomalies leads to an average of 0, while
the median is slightly negative following the skewed histograms in Fig. 5. Precipitation plots in Fig. 6a present the widest
distribution during summer (DJF), followed by autumn (MAM) and spring (SON). For each season, boxplot lower and upper
whiskers stand for the 5$^{th}$ and 95$^{th}$ percentiles; values outside whiskers (i.e., below the 5$^{th}$ percentile or above the 95$^{th}$ percentile)
represent the outliers. The figure shows that the more extreme dry events can happen during summer and autumn, with outliers

reaching -100 mm. By contrast, during winter (JJA) most of the values are found near 0 mm with small deviations: outliers
around -25 mm indicate that the events in this region are not necessarily extreme during winter.

Boxplots for soil moisture in Fig. 6b show that seasonal distributions are more uniform, probably due to their lower variability
and lower range values than for precipitation. Interestingly, the outliers have the largest magnitudes during autumn (MAM),

reaching deviations between -20 and -30 mm. This result is consistent with a delay with respect to precipitation, which showed
the most extreme cases during summer (DJF). In fact, the delay also results in that soil moisture also exhibits more extreme
cases during winter (JJA) again following the large values for precipitation during autumn (MAM).

### 3.1.4 Drought duration

Drought duration is defined as the number of months that a given drought index (SPI and SSI) exceeds a certain threshold, $X_i$.

For both SPI and SSI, the value $X_1 = -0.5$ identifies mild to extreme droughts, while using $X_2 = -1$ detects moderate to extreme
droughts. Figure 7 shows the SPI and SSI frequency of droughts inside the Core Crop Region regarding different durations of
the events, expressed in months. Each histogram presents the number of events for each duration, hinting to different types of
droughts.

All histograms in Fig. 7 present a common pattern with a higher frequency for short-term events (1-3 months). The frequency
(or the number of cases) declines as drought duration increases. These results suggest that long-term droughts, particularly





beyond seven months, are uncommon inside the Core Crop Region. Table 2 presents the percentages of drought occurrence for short-term droughts and more prolonged than three months events as characterized by SPIs and SSI at time scales of 3 and 6 months. The SPI indices have more ability to identify short-lived droughts than the standardized index based on soil moisture. In contrast, SSI seems a better fit to detect longer droughts (see Table 2). In summary, short-term droughts are better represented by an index like SPI, with higher variability and short time scale. Long-term drought events are more easily detected with an index of lower variability and a higher time scale.

### 3.1.5 Severity and spatial extent of droughts

Drought duration and magnitude are important to describe droughts, as well. It is central to have a measure of severity and spatial extent of the drought. Severity can be defined as the product between the drought duration and drought magnitude. A drought's spatial extent refers to the area that exceeds a certain threshold (e.g., $X_2$), and it is expressed as a percentage of the total Core Crop Region. Figure 8 presents the time series of severity and spatial extent computed from SPIs and SSIs for the Core Crop Region. According to Figs. 8a, b, the most severe droughts occurred during 1988/89, 1995/96, 2008/2009, and the last one during 2017/2018, consistently with the analysis in Fig 4. Series of severity are negative because they result from a product of a negative drought magnitude (defined by using a negative threshold like $X_2$) and a positive duration. Severity indexes seem to be greater in magnitude (more negative) when computed from 6-month time scales (SPI6 and SSI6), which is due to a lesser index variation as the time aggregation of the index increases.

The Core Crop Region extends over 500,000 km$^2$ in the center of Argentina (shown in Fig. 1). Inside this area, Fig. 8c suggests that the more severe droughts were also the ones with a greater spatial extent. Further, for every severe event, the SPI time series indicate that droughts are extended around 80 to 90% of the core crop region, increasing the impacts of these events on the main activities of the region. Even droughts that are not quantitatively as severe can spread almost in equal proportions as those severe ones; for instance, the events that have occurred in 1988, 1995, and 2007 (see magenta lines in Fig. 8c). The soil moisture lower variability results in similar time series of SSI3 and SSI6 (Figs. 8b, d). Therefore, droughts in the Core Crop Region are detected more easily when using SSI instead of SPI. This is the case of drought events occurring in 1988, 1997, 2007, 2009, and 2017 that spread over 80% to 90% of the region (Fig. 8d).

### 3.2 Crop yields in the core crop region

Due to the importance for regional economies, it is always of interest to stress the negative impact of droughts on crops. Changes in crop yield, defined as crop production per unit area (kg ha$^{-1}$), reflect not only the effects of climate variability but also technological and biotechnological advances, usually in the form of a positive nonlinear trend. This can be noticed in Fig. 9, which presents the 1969-2018 area-averaged time series of corn, wheat, and soybean yields for the provinces of Santa Fe and Córdoba that cover most of the Core Crop Region (see Fig. 1d). The wheat and soybean trends show a significant change around the mid-1990s, whereas, for corn, a change occurred earlier in the late 1980s. On average, wheat and soybeans yields





increased from 1,000 to 3,000 kg ha$^{-1}$ (Figs. 9a and 9c) while corn yield increased from 3,000 to almost 8,000 kg ha$^{-1}$ (Fig.

9b). As stated, at least the majority of the increases may be due to advances in the production process. These trends should be removed when examining the crop yield variability and its relation to droughts. Crop yield time series were fitted with a cubic polynomial trend (see dotted lines in Figs. 9a-c). Then, the trends were subtracted from the original series, leaving the shorter-term variability (see Figs. 9 d-f). Detrended time series of one or more crop yields exhibit the largest negative anomalies concurrently with the most severe droughts identified by SPI6 and soil moisture anomalies (Figs. 4c and 4d) recorded in

1988/1989, 1995/1996, 2008/2009, and 2017/18 (Figs. 9d-f). Note that not all crops are affected equally by drought as slight differences in the onset of the drought and the crops' critical growth periods that may affect them differently.

Drought severity values are good indicators of crop yield losses, as noted above. Large negative anomalies of corn and soybean yields (Figs. 9e and 9f) are consistent with the largest severity estimations during the drought events in 1988/1989, 1995/1996

and 2017/18 (Fig. 8), when severity acquired values around -8 for SSI3 and SSI6 through crops critical growth period (see symbols in Figs. 9e and 9f), which constitute extreme negative values. Similarly, drought severity values in 2009 (see symbols in Fig. 9d) were related to a large impact in wheat production, particularly for Santa Fe Province (see blue line in Fig. 9d).

Some resemblance in detrended series of corn and soybean are found between Figs. 9e and 9f. Although anomaly values are

somewhat different, the time series behavior seems to be in phase with each crop's yield. Increases and decreases in production take place nearly at the same time, unlike the behavior with wheat (Fig. 9d). In particular, both crops present major yield losses approximately in the years of the main droughts (Figs. 9e and 9f). This could be related to the one month overlap during both sensitive periods, as the two crops have January as a common month during their critical growth in summer.

Drought severity values and crop production have a better representation with detrended series of corn and soybean yields, and particularly a good fit is observed for 1988/1989 and 1995/1996 events. Conversely, from 1998 to 2007, no severe events occurred. For those years, severity values are somewhat constant and close to -4 for all the indices (Figs. 9d-f); on the other hand, crop production ranges from neutral to positive values, with a maximum for corn in Córdoba. This could imply that this level of severity during their sensitive months is not enough to affect productivity for those crops. The analysis of indicators

capacity to represent downs in crop production with proper severity values shows that soil moisture indexes are more consistent and seem to outperform precipitation indices in that matter. SSI severity values are consistently higher in magnitude than SPI (as seen in Figs. 8a and 8b) and seem to carry out better in picturing yield's declines. Therefore, SSI constitutes a better indicator than SPI to relate crop yield losses and negative impacts due to droughts.

A general comparison among the crops, in terms of the magnitude of production declines due to major drought events, proves that corn is more sensitive to droughts than wheat and soybean. Of all the crops, corn remains the one that is more affected when water availability in the soil is compromised: The losses in production may reach up to 1500 kg ha$^{-1}$. For wheat and



soybean, the losses are more alike and between 500 to 1000 kg ha$^{-1}$. As a result of this behavior, corn exposes as the most sensitive crop of the three to water deficit (or excess) in terms of crop productivity.

**4 Discussion and conclusions**

This study documents drought in Argentina's Core Crop Region, where the production of wheat, corn, and soybean is the most abundant. The drought properties that were examined include magnitude, duration, severity, and areal extension. The analysis was completed by examining the relationship between drought properties and crop yields. The investigation is based on the analysis of anomalies of precipitation and soil moisture, and their derived indices SPI and SSI, respectively, at different time

scales. Droughts were identified as the events that have a water shortage exceeding one standard deviation or more of the mean values. The requirement was slightly relaxed for estimating the duration of events, considering water deficits that depart at least half a standard deviation. The analysis was performed at different time scales: all-months, seasonal, and on a monthly basis for each crop's critical growing months. Crop yields have increased through the years in part due to more beneficial climate conditions, but more importantly thanks to agro-technological advancements. We inspected drought impacts on crop

yield after removing those trends.

Our results indicate that the probability of drought occurrence depends on whether SPI or P anomalies are used, as the standardized nature and time aggregation of the SPI index tends to emphasize longer time scales. Short-term droughts are more easily detected when using an index with higher variability and short time scale. For this reason, short term drought-prone

regions and their relation to seasonality within the core crop region are better identified using precipitation anomalies rather than standardized indices. The presence of long-term events is more readily recognized with an index of lower variability and at lower time scales.

Spatial patterns of drought probability for all times considered do not show clear features. The probability of occurrence of

drought in northeastern Argentina ranges between 12-18% depending on the variable (P, SM) and location, with the larger values found towards the eastern/northeastern sector of the Core Crop Region. The probability of occurrence, based on soil moisture anomalies, shows a second area of slightly high values for the semi-arid climate towards the western portion of the Core Crop Region. During summer, droughts affect the corn and soybean production mainly towards the west and center of the Core Crop Region.


Soil moisture acts as a temporal filter in the sense that it smooths out the highly variable precipitation resulting in a lower frequency signal. Similarly, the variability of the SPI time series is reduced when the time scale increases from three months to six. Frequency analysis for different durations indicates that short term droughts are more common than long term droughts. Our findings show that values accumulated for 1-3 months account for about 78-88% of the events when depending of the



threshold and variable considered. A few can extend up to one year and even fewer even longer. However, if a multiyear drought experienced breaks, each period would be considered a separate case. These results are consistent with drought frequency values found by Minetti et al. (2007), who reported that similar 1-3 months events account for 90% of the cases in the Argentine Humid Pampa. Small differences could be related to the use of different indexes and thresholds in the definition of drought. In general, long-term drought events are more easily detected with an index of lower variability, like SSI, and a

higher time scale.

Each crop type has its fastest growth during short periods sometime in spring or summer. These critical periods are susceptible to water availability so that even a short duration dry event, if concurrent with the critical growth period, may have a large impact on the crop performance. Large drought severity values taking place during sensitive months will result in significant

crop yield losses. This is important because if a high severity drought event is identified and quantified during crop-sensitive months, it will be an indication that yields of that crop can be expected to be lower, potentially experimenting significant economic losses on the next campaign.

Many previous studies (e.g., Barrucand et al., 2007; Vargas et al., 2011; Lovino et al., 2018a, b) detected a positive increment

in the trend of rainfall. Consistent with these results, Lovino et al. (2018a, b) suggested that short-term variability (2.5- to 4-year periods) in precipitation exhibits a large increase in amplitude after 2000, contributing to a higher occurrence of wet and dry events in the first two decades of the 21$^{\text{th}}$ Century, and in particular favoring interannual drought episodes. The time evolution of the different variables presents an intriguing behavior with a notable change in interannual variability starting around 1998.


Argentine agriculture has benefited from the increased use of fertilizers, agrochemicals, and the management of genetically modified crops, leading to important positive trends in crop yields. Detrended yield time series facilitate their comparison to climate variations. Our study shows that unlike wheat and soybean, corn seems to be the most sensitive crop to water deficit (or excess) in terms of crop productivity in the Core Crop Region. As a note of caution following Butler and Huybers (2013),

corn production may be affected not only by water availability but also by temperature and geographical adaptation, two features whose analysis is outside the scope of this study. During crops sensitive months, higher absolute values of severity seem to be coincidental in time with mayor yield losses during mayor drought events. This was the case for summer crops, and soil moisture derived indices during 1988/89, 1995/96, and 2017/18 events. In consequence, timing is very important because if a drought with high severity could be identified and quantified during crop-sensitive months; it could be an

indication that yields of that crop would be negative with potential major harvesting losses on the next campaign.




**Data availability**

CPC precipitation and GLDAS soil moisture data sets are available at NWS, CPC, https://doi.org/10.1029/2007JD009132, Chen et al., 2008; and GLDAS, https://doi.org/10.5067/9SQ1B3ZXP2C5, Beaudoing and Rodell, 2019; https://doi.org/10.5067/SXAVCZFAQLNO, Beaudoing and Rodell, 2020. Crop yields data sets are provided by MAGyP, available at http://datosestimaciones.magyp.gob.ar/reportes.php?reporte=Estimaciones.

**Author contribution**

Author and co-authors contributed to the design and implementation of the research, to the analysis of the results and to the writing of the manuscript.

**Competing interests**

The authors declare that they have no conflict of interest.

**Acknowledgments**

This research was carried out with the support of Projects CRN3035 and CRN3095 of the Inter-American Institute for Global Change Research (IAI), which is supported by the US National Science Foundation. NOAA Grant NA19NES4320002 (Cooperative Institute for Satellite Earth System Studies, CISESS) and Project IO-2017-00254 of the Ministry of Science, Technology, and Productive Innovation of Santa Fe (Argentina) are also acknowledged.

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






| | Pre Anomalies | SM Anomalies | SPI3 | SPI6 |
|---|---|---|---|---|
| Trend | 13.7 | - | 3.2 | 5.5 |
| Quasi-cycle, T ~ 10 yr | - | 25.4 | 13.9 | 22 |
| Quasi-cycle, T ~ 3.4 yr | 17.3 | - | - | - |
| Quasi-cycle, T ~ 2.3 yr | 24 | 11.6 | 7.8 | 13 |

**Table 1: Percentage of variance explained by the dominant modes of interannual variability detected using SSA with a window length of 120 months. Series of precipitation anomalies, soil moisture anomalies, SPI3 and SPI6 from January 1979 to December 2018 in Core Crop Region.**





| | Duration [1-3 months] | | Duration [ >3 months] | |
| --- | --- | --- | --- | --- |
| | $(X_1)$ | $(X_2)$ | $(X_1)$ | $(X_2)$ |
| SPI3 | 77.8 | 88.2 | 22.2 | 11.8 |
| SPI6 | 68.7 | 76.7 | 31.3 | 23.3 |
| SSI3 | 46.7 | 49.9 | 53.2 | 50.1 |
| SSI6 | 38.5 | 44.0 | 61.5 | 56.0 |

**Table 2: Frequency of droughts for different durations, expressed as a percentage of the total events in Core Crop Region, from indices SPIs and SSIs. Droughts were detected using threshold $X_1$ (one half standard deviation) and $X_2$ (one standard deviation) from January 1979 – December 2018. Duration of events were grouped into short-term [1-3 months] and longer droughts [>3 months].**



**Figure 1: (a) Map of southern part of South America with topographic levels and country names. Extension of the Core Crop Region (highlighted with a dark brown rectangle) defined as the most productive region for the three crops combined in Argentina with names of the provinces. Intensity of production in [tn] for each crop taken from seasons 2010/11-20017/18 for soybeans and corn, and from 2010/11-2018/19 for wheat. Production lower intensity in orange to higher in dark green: (b) for soybean, (c) for wheat, and (d) for corn. Also, each panel shows crop's development cycle for Argentina where grain-filling and flowering period represent the most sensitive period: as (Oct-Nov) for wheat, (Dec-Jan) for corn, and (Jan-Feb) for soybean (MAGPyA).**



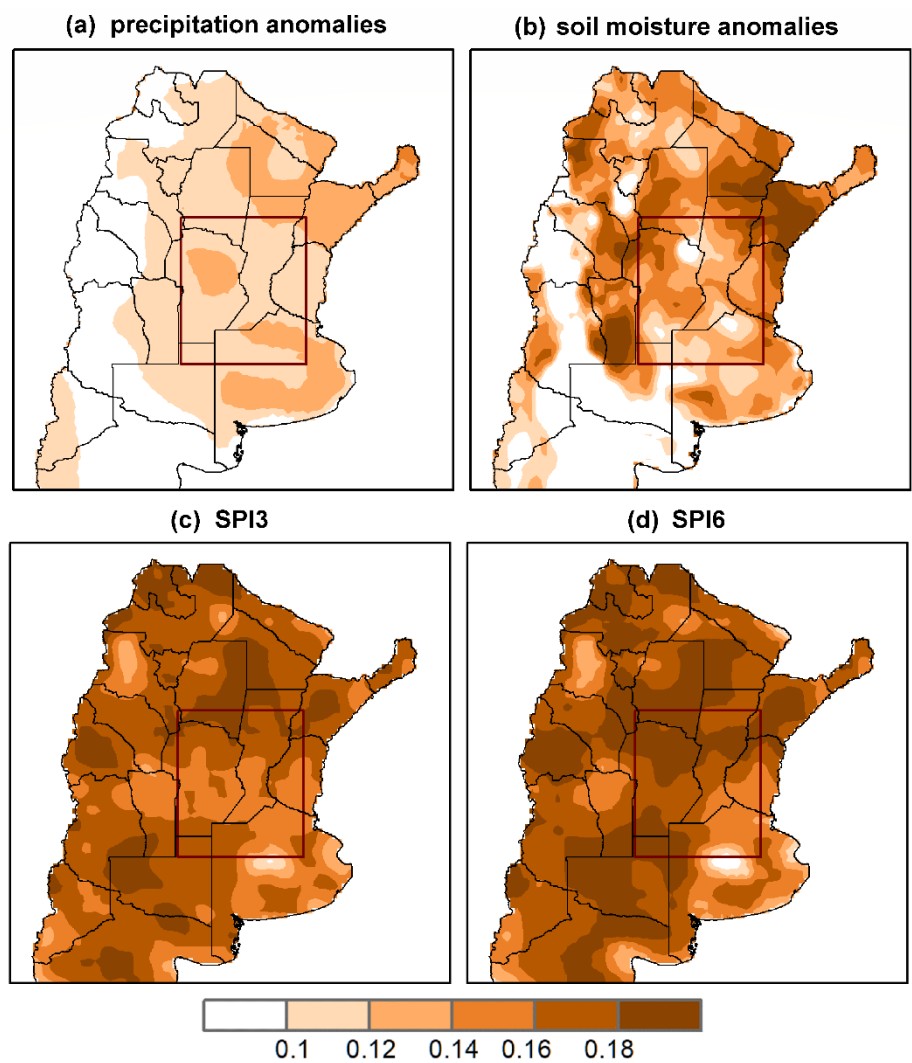

**Figure 2: Probability of drought events below one standard deviation from January 1979 to December 2018, according to different variables: (a) Precipitation anomaly, (b) Soil moisture anomaly, (c) SPI3 and (d) SPI6.**



**Figure 3: Probability of drought events below one standard deviation accounted during each crop critical growing months from January 1979 to December 2018. For SPI3: (a) wheat during (Oct-Nov), (b) corn during (Dec-Jan), (c) soybean during (Jan-Feb). Panels (d), (e), and (f) showing the same for SPI6.**



**Figure 4: Areal-averaged time series from January 1979 to December 2018 for (a) precipitation anomalies, (b) SPI3, (c) SPI6, and (d) soil moisture anomalies in the Core Crop Region. The dominant modes of interannual variability are plotted in full lines.**




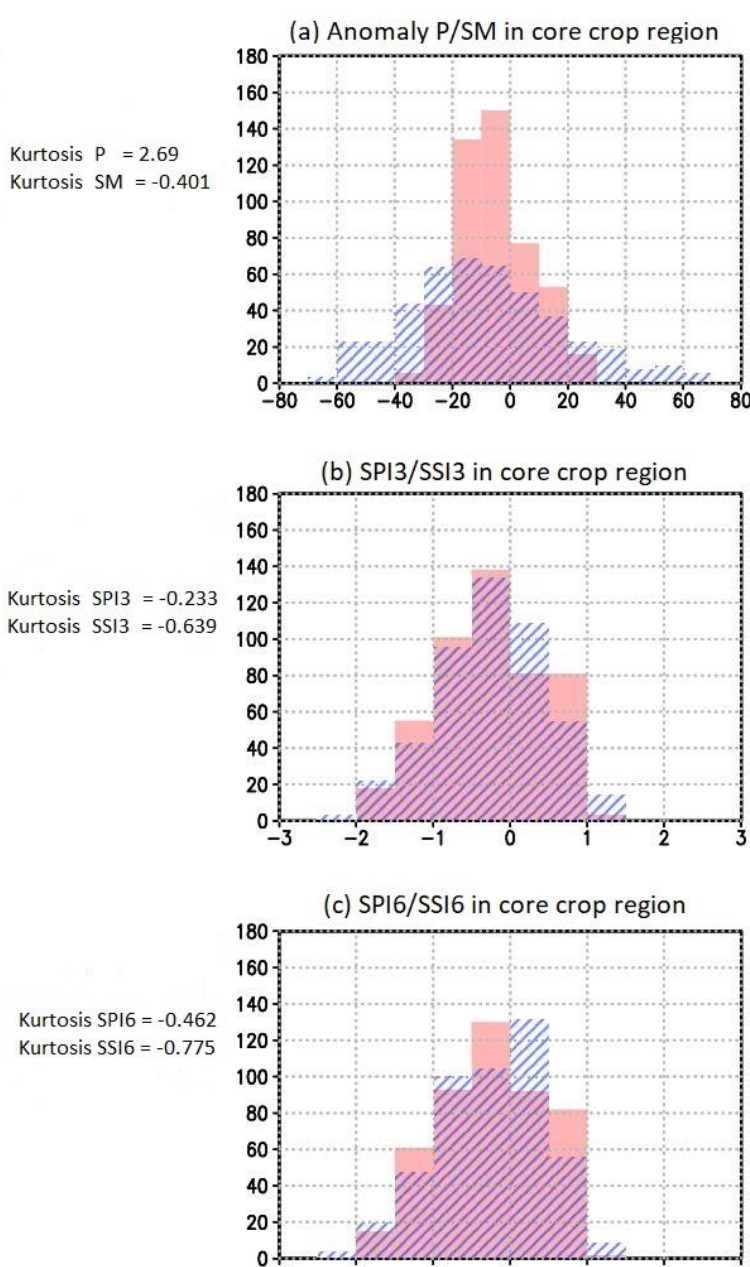


**Figure 5: Frequency histograms in Core Crop Region from January 1979 to December 2018, and Kurtosis values for: (a) Precipitation anomaly (blue-hatched) and Soil moisture anomaly (light red), (b) SPI3 (blue-hatched) and SSI3 (light red), (c) SPI6 (blue-hatched) and SSI6 (light red).**








**Figure 6: Boxplots from January 1979 to December 2018 seasonal averaged time series inside Core Crop Region for: (a) Precipitation anomaly (light blue), (b) Soil moisture anomaly (light red).**


**Figure 7: Histograms of droughts for different duration in months from January 1979 to December 2018 in Core Crop Region, represented by (a) SPI3, (b) SSI3, (c) SPI6, and (d) SSI6. Colored bars indicate mild to extreme droughts which values are lesser than $X_1 = -0.5$. Hatched bars in all the panels indicate moderate to extreme droughts which values are lesser than $X_2 = -1$.**




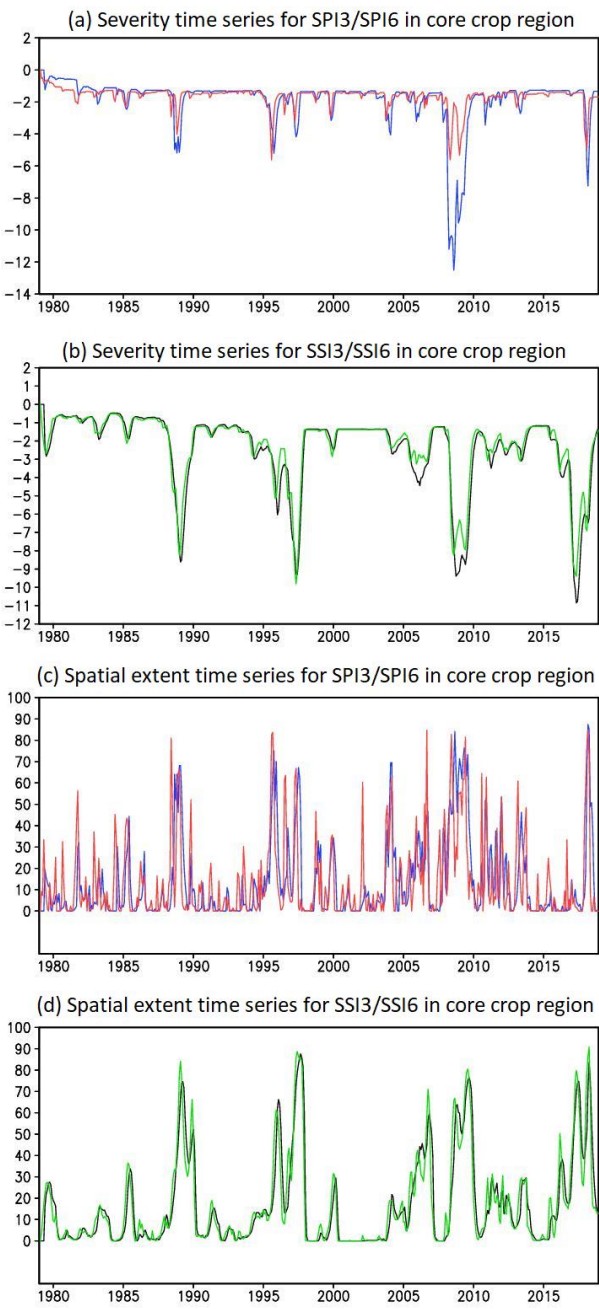

**Figure 8: Average time series of drought severity from January 1979 to December 2018 for events below X₂ in Core Crop Region, for: (a) indexes SPI3 (red) and SPI6 (blue), (b) as Panel (a) for SSI3 (green) and SSI6 (black). Average time series of drought spatial extent from January 1979 to December 201,8 expressed as a percentage of total Core Crop Region, for events below X₂, for: (c) indexes SPI3 (red) and SPI6 (blue), (d) as Panel (c) for SSI3 (green) and SSI6 (black).**




**Figure 9: Panels a-c, show the time series of the area-averaged annual crop yield over the provinces of Santa Fe (blue lines) and Córdoba (orange lines) from 1969 to 2018 for: (a) wheat, (b) corn, and (c) soybean. Cubic polynomial trends of each province crop yield time series are in dot line. Panels d-f, presents the detrended time series together with minimum severity index values during each crop critical growing months: (d) Minimum severity index indicated for SPI3 (blue crosses), SPI6 (black crosses), SSI3 (green crosses) and SSI6 (magenta crosses) for months (ON). Rest of the panels show the same as (d) but for summer sensitive crops: (e) for corn and (f) for soybean, minimum severity values during (DJF) are represented for SPI3 (blue triangles), SPI6 (black triangles), SSI3 (green circles) and SSI6 (magenta circles).**