# Peer review of "Characteristics of droughts in Argentina's Core Crop Region"

_Hydrology and Earth System Sciences, 2020_

## Referee Comment (RC1) · Anonymous Referee #1 · 6 Oct 2020

The main idea of the paper is very interesting, specifically because of the usefulness for the Agricultural Sector and Argentinean GDP. The paper is well organized and the ideas are clearly explained.

After reading the paper, the idea of understanding how the droughts affect the crops yield is well explained and, specifically, how the different variables related to droughts affect different type of crops.

Some specific aspects about your work:

* In equation 1 and 2, when describing "xi", it would be better if you consider the "i" as a subscript.

* I think that there is an error in line 205: "that is higher than for precipitation" should

be " that is higher for precipitation".

* In Figure 3 the color bar is not clear enough. The top value (0.18) is reached in an important fraction of the area, and the impression is that higher values can also be possible. Maybe changing the color bar and allowing a higher limit can improve that figure. Some of your comments are related to the 0.18 probability, but that can change if a different color bar is used.

* When you estimate SPI3 and SPI6 for the critical months and analyze the temporal evolution of drought frequency. Is there any overlapping between these months and the estimation of the SPI-SSI values? How does that overlapping affect your conclusions?

* In line 230 you mention that SM acts as a physical filter, but there is no mention on the mechanisms. Maybe exploring the physical processes involved should be incorporated and analyzed.

* The Cordoba Province is written as Córdoba and Cordoba (with and without accent).

* In Figure 8, the inclusion of a legend will facilitate the understanding of the meaning of each line.

* Figure 9 shows the detrended crop yield and the SPI-SSI indexes and your conclusions say that lower SPI-SSI values implies crop yield losses, but that is not the case for every year. Is there any explanation?. Why are only some years shown? Maybe a direct comparison between the SPI-SSI indexes and the crops yields (instead of a time series), considering all the years, can be more descriptive. Your conclusions depend on this figure and should be clearer the relation between SPI-SSi and crop-yields and explain years without a reasonable results, like the 2004-2009 period.

---

## Referee Comment (RC2) · Anonymous Referee #2 · 21 Oct 2020

The authors' present an interesting manuscript which discuss and deal with a climatology of different drought properties such as magnitude, frequency at different time scales, duration, and severity in central Argentina. Drought conditions along 40 years of rainfall and soil moisture records and the related standardized indices (SPI and SSI) are analysed. They conclude that most droughts tend to occur for periods shorter than three months, but a few can extend up to one year and fewer even longer while in the the core crop region, corn yield is the most sensitive crop to dry conditions. The topic of the paper fit well in HESS and is of regional interest to HESS readers. While the work and findings are interesting, there are some issues that need to be clarified.

General comments.

While I understand what the authors are trying to do, as it stands now, the manuscript

appears to be a collection of statistical analyses related in some way to drought issues, but no clear objective or story is followed. In particular, what is the main goal of the paper? Is it aimed at developing a monitoring system, to provide information on the link between the indicators and crop yield or just provide a climatology? It seems that the research is focused on the main 3 crops but at the end only a qualitative comparison with yields is shown. I recommend to focus and tailor the discussion in one aspect, in particular, the link with crop yields (see some comments below).

Abstract: The abstract could be improved, it should be rewritten focussing on the main goals of the manuscript and trying to avoid general statements like: "It is of interest to assess the relationship between those properties and the crop yields." Or "As relevant as the drought duration is its timing and severity".

Introduction: In the last two paragraphs of the intro it would be better to focus on the goals of this paper and how this advances the knowledge in this specific subject. Details on drought indicators should be explained in the methods section.

Methods section: the description of yield data is somehow misplaced and cuts the logical flow of the description of the meteorological variables. Consider placing it at the end of the section after the description of the standardized indicators.

I'm curious to know why this methodology was selected (eqs. 1 and 2). Does this approach perform better than a parametric approach? I guess that the length of the records (40 years) shouldn't be a problem to fit a parametric distribution. Please motivate this decision and provide a more critical discussion on the methodologies adopted, including advantages and possible shortcomings.

Section 2.3: This is a key section that needs to be strengthened. The drought definition (in the first paragraph) is quite general and vague and it is not clear how this is implemented in the following description. Please be explicit here, e.g. "in our framework a drought starts when our drought indicators are below 1 sd and last until it reaches (threshold) again... ". Moreover, it seems that only one month with negative values of

the indicators defines a drought. This is not the most accurate definition of droughts, which need some time to become established, in particular with the threshold chosen (-0.5). However there are some cases where droughts can develop quickly, e.g. flash droughts and lead to large impacts, but not sure this is reflected here. Please, again, motivate clearly your drought definition. This is key to follow the results.

The use of precipitation anomalies is not the best approach to make a regional assessment. How do you compute those? I guess P-P*. This procedure could give misleading results, as the anomalies are not comparable between regions or even different months/seasons. Let's suppose you have one locality with observed 50 mm in one month where the monthly mean is 100 mm, then the anomaly is -50 mm, then for another locality you have observed 5 mm but the monthly mean is 55mm, here your anomalies are again -50 mm. How these deficits are comparable? How do you include those in the areal average? If the goal is to capture the high frequency variations a more suitable indicator would be the percent of Normal precipitation or the SPI-1. Please, for further discussion on that matter refer to the WMO-GWP handbook of drought indicators and indices and references in there.

Section 3.1.1. There is an issue with the results in this section, probably due to the lack of drought definition (particularly Figs 2 and 3). The authors compute the probability of occurrence over standardised indicators which means that the probability to be below -1 sd should be 0.1587 if these indicators follow a perfect normal distribution. I'm afraid that this is what is shown here with the spatial variability due to some statistical instability. Here a more meaningful series of maps could be the return period of droughts or even the number of drought events per pixel.

Section 3.1.3. Same issue here as in section 3.1.1. Maybe I'm missing something important here, but what is the added value of showing a histogram for a set of standardised variables?. Even if these represent a regional average, each single point should follow a normal distribution with mean=0 and sd=1, any deviations from this means some problems with the fitting or the input variables (marked dry season, many

zeros, etc.). Please clarify, adapt your description towards potential problems in the fitting or add additional relevant information.

Section 3.2: There are many studies that relates climate variables and droughts to losses on crop yields in the region (Holzman et al 2014; Podesta et al. 2009; Scian and Bouza 2005; Seiler et al 2007 to cite but a few). This section presents a qualitative description of negative impacts of droughts on crops. Please, clarify how these results advance the literature on the topic in the region or how they are relevant in the context of this study. I believe that the data presented are valuable and could help to answer some questions proposed here, such as what indicator is most appropriate to predict crop losses. For example, a correlation analysis/ contingency table, etc. between the different indicators (and specific dates sensitive to the crops) and the crop losses would make this section much more relevant. This could support some affirmations like the ones on lines 396-402. At the moment the highly sensitive months are presented a-priori.

The affirmation that corn crop is the most sensitive to water deficits is due to the fact that it is the crop with higher yield/ha (more than 2x to than the other according to lines 323-324). Relative losses related to their average yield values would make their values comparable or even could be greater for the other two crops. As you mention lately corn is also highly sensitive to heat waves as well. Compound events could intensify these losses, but this is not discussed here.

Particular comments

Line 17: "However, if a multiyear drought experienced breaks, each period would be considered a separate case" this is not clear, please consider rephrasing or even removing this from the abstract.

Lines 19-20: "Even short dry spells may have large impacts if they occur at the time of the critical growth period of a given crop" this is not discussed or demonstrated in the manuscript. Please, if this is a general statement, consider removing it from the

abstract or perform more analysis that can support this affirmation.

Lines 21-22: This also seems a general and rather speculative statement. Consider removing it from this section or perform more analysis that can support this affirmation (see my comment for section 3.2).

Lines 122-126: The performance of a model in the representation of soil moisture is not trivial. Here, there is no mention on how the GLDAS performs in the region. A formal validation is not necessary but at least some references describing its validity in the region are needed. See for instance, Spennemann et al., 2015 & 2020.

Line 47: This only apply to dry events; wet events has a completely different dynamic. Please, clarify.

Line 69: Agriculture will be only one sector affected. Droughts can affect almost any compartment of economy and ecosystems.

Line 104: the sentence between [. . .] should be removed of placed elsewhere.

Line 203-205: how the soil moisture can be affected by soil degradation and desertification? Is this modelled in GLDAS?

Line 207: The SPI-6 could be correlated with hydrological droughts (probably longer aggregation periods can do it better) but it is still a meteorological indicator as relies only on precipitation. It is not accurate to attribute the representation of hydrological droughts to it. Please consider rephrasing this sentence.

Line 216-217: How the maps in Fig 3 can be interpreted as the temporal evolution of drought frequency?

Lines 314-315: "Therefore, droughts in the Core Crop Region are detected more easily when using SSI instead of SPI." It is not possible to benchmark drought detection by just comparing two drought indicators. Just because one indicator covers more area or is more severe, etc doesn't mean that is the best performing indicator. Instead, I

would argue that the best indicator should be the one that better represents the specific sectorial impacts, in this case, crop production.

References

Holzman, M. E., Rivas, R., & Piccolo, M. C. (2014). Estimating soil moisture and the relationship with crop yield using surface temperature and vegetation index. International Journal of Applied Earth Observation and Geoinformation, 28, 181-192.

Podestá, G., Bert, F., Rajagopalan, B., Apipattanavis, S., Laciana, C., Weber, E., ... & Menendez, A. (2009). Decadal climate variability in the Argentine Pampas: regional impacts of plausible climate scenarios on agricultural systems. Climate Research, 40(2-3), 199-210.

Scian, B. V., & Bouza, M. E. (2005). Environmental variables related to wheat yields in the semiarid pampa region of Argentina. Journal of arid environments, 61(4), 669-679.

Seiler, R. A., Kogan, F., Wei, G., & Vinocur, M. (2007). Seasonal and interannual responses of the vegetation and production of crops in Cordoba–Argentina assessed by AVHRR derived vegetation indices. Advances in Space Research, 39(1), 88-94.

Spennemann, P. C., Rivera, J. A., Saulo, A. C., & Penalba, O. C. (2015). A comparison of GLDAS soil moisture anomalies against standardized precipitation index and multisatellite estimations over South America. Journal of Hydrometeorology, 16(1), 158-171.

Spennemann, P. C., Fernández-Long, M. E., Gattinoni, N. N., Cammalleri, C., & Naumann, G. (2020). Soil moisture evaluation over the Argentine Pampas using models, satellite estimations and in-situ measurements. Journal of Hydrology: Regional Studies, 31, 100723.

WMO, G.; GWP, G. Handbook of Drought Indicators and Indices. Geneva: World Meteorological Organization (WMO) and Global Water Partnership (GWP), 2016.

---

## Author Comment (AC1) · 19 Dec 2020

Responses to Reviewer 1

The author would like to thank the reviewer for the constructive and thoughtful comments and suggestions which led to substantial improvements in the revised version of the manuscript.

Attached please find our response.

Sincerely, Leandro Sgroi

Please also note the supplement to this comment:

[Figure]

Please also note the supplement to this comment:
https://hess.copernicus.org/preprints/hess-2020-236/hess-2020-236-AC1-supplement.pdf

―――――――――――――――――

[Figure]

**Supplement:**

**Responses to Reviewer 1**

The author would like to thank the reviewer for the constructive and thoughtful comments and suggestions which led to substantial improvements in the revised version of the manuscript.

The reviewer's comments are marked in bold and italic. The responses follow in indented text.

**Some specific aspects about your work:**

**1. In equation 1 and 2, when describing "xi", it would be better if you consider the "i" as a subscript.**

> Corrected in the revised manuscript.

**2. I think that there is an error in line 205: "that is higher than for precipitation" should be "that is higher for precipitation".**

> We realize that the sentence was confusing. In the revised version of the manuscript, the variable "precipitation anomalies" has been replaced by SPI1.
>
> The entire paragraph has been rewritten. It now states:
>
> *"Figure 2 presents the spatial distribution of drought's occurrence percentage for northern Argentina as characterized by SPI1, SPI3, SPI6, and soil moisture anomalies lower than one time their corresponding standard deviations. The occurrence of drought in northeastern Argentina ranges between 12 and 14% for SPI1 (Fig. 2a), while drought percentages seem to increase up to 18% as characterized by SPI3/6 in the same region (Figs. 2b and 2c). Soil moisture anomalies show that droughts are distributed mainly in the north of Argentina, with percentage values ranging between 16% and 18%. Droughts, as characterized by SPI1/3/6 (Figs. 2a-c), reveal a homogeneous spatial distribution and increasing percentages as with the time scale of the indicator. In contrast, the spatial pattern of soil moisture anomalies shows a decrease in drought percentages for arid regions (Fig. 2d). Inside the Core Crop Region, droughts are more frequent towards the north, with percentages from 14% to 16% for SPI1/3 (see Figs. 2a-b). Fig. 2d indicates that the percentage of droughts, as represented by SPI6, is 18% towards the region's north and southwest. Conversely, drought presence declines towards southeastern CCR as all SPI, and SM show percentages descend to 12% (Figs. 2a-d)."*

**3. In Figure 3 the color bar is not clear enough. The top value (0.18) is reached in an important fraction of the area, and the impression is that higher values can also be possible. Maybe changing the color bar and allowing a higher limit can improve that figure. Some of your comments are related to the 0.18 probability, but that can change if a different color bar is used.**

> Probability was not the proper expression for Figures 2 and 3. Instead, we now use percentages of drought occurrence. Figures 2 and 3 have been redone accordingly (see below).

The drought's occurrence percentage does not exceed 20% in any of the cases. The largest values in Figure 3 are in the range 18 % to 20%. As an example, we present below the class distribution histogram for SPI6 wheat critical months (Fig. 3d). The rest of the maps in Figure 3 present very similar histograms.

To clarify this issue, we have adjusted the color bars in Figs. 2 and 3 by adding the upper maximum limit of 20%.

[Figure]

**4. When you estimate SPI3 and SPI6 for the critical months and analyze the temporal evolution of drought frequency. Is there any overlapping between these months and the estimation of the SPI-SSI values? How does that overlapping affect your conclusions?**

Critical months refer to the specific periods when the crops are most sensitive to ambient conditions. These periods do not necessarily cover the predetermined seasons, e.g., SON or DJF.

Therefore, their analysis is aimed at evaluating the potential of droughts to affect crop yields during those specific periods. Even though there is an overlap with the 3- and 6-months' time scale for indicators, it does not affect any of the conclusions presented in our study.

**5. In line 230 you mention that SM acts as a physical filter, but there is no mention on the mechanisms. Maybe exploring the physical processes involved should be incorporated and analyzed.**

In the framework of the surface water budget, soil moisture is estimated as a difference between incoming precipitation and outgoing runoff and evapotranspiration (these are the main terms). Different mechanisms govern the behavior of each of the components as well as how they interact in a temporal scale. When precipitation enters the system, its signal will propagate in time. The fundamental concept was discussed by Changnon (1987) and McNab (1989) using the schematic shown on the side. The schematic shows a time delay between variables and a smoothing of the curves.

[Figure]

Part of the precipitation that reaches the ground will evaporate from the surface, another part will turn into surface runoff, and the rest will infiltrate into the ground. Some of the infiltrated water may go back to the atmosphere through vegetation transpiration of water extracted by roots. Another portion may percolate to groundwater, and the rest will be stored as soil moisture (Changnon, 1987; Van Loon, 2015).

The time it takes for the precipitated water to infiltrate the soil and through deeper layers has s dampening or smoothing effect that Entekhabi et al. (2006) described as a low-pass filter.

We have added a discussion of this process to the revised manuscript. The analysis if evapotranspiration, runoff, while attractive, would require a different approach (based in land surface models) that is out of the scope of our study.

**6. The Cordoba Province is written as Córdoba and Cordoba (with and without accent).**

The proper spelling is with an accent. We have corrected it throughout the manuscript.

**7. In Figure 8, the inclusion of a legend will facilitate the understanding of the meaning of each line.**

Thanks for pointing it out. A new legend has been included in the figure.

**8. Figure 9 shows the detrended crop yield and the SPI-SSI indexes and your conclusions say that (a) lower SPI-SSI values implies crop yield losses, but that is not the case for every year. Is there any explanation? (b) Why are only some years**

*shown? (c) Maybe a direct comparison between the SPI-SSI indexes and the crops yields (instead of a time series), considering all the years, can be more descriptive. (d) Your conclusions depend on this figure and should be clearer the relation between SPI-SSi and crop-yields and explain years without reasonable results, like the 2004-2009 period.*

(a)      Climate conditions (as those given by SPI or SSI) affect crop yields, but they are not the only factors at play. Non-climatic factors like seed quality, different use of fertilizers, sawing or harvesting dates are a few of the possible effects on crop yields. This issue has been clarified in the revised manuscript.

A better way to support our results is by computing correlations between the drought indices and crop yields. The Table 3 below (to be included in the manuscript) shows that indeed there is a relation between indices (SPI3/6 and SSI3/6) and crop yields. Table 3 shows that the shorter-scale indicators (SPI3 and SSI3) achieve a better correlation with crop yields than the larger-scale indicators (SPI6 and SSI6). Then, SPI3 and SSI3 result most appropriate to predict crop yield losses. The relation between detrended crop yields and SPI3 is now presented in the revised Figure 9 (see below).

The results suggest a sensitive relation between summer crops (corn and soybean) and deficits in precipitation and soil moisture during both crops' critical periods. Of the three crops, wheat yields have the lowest correlations with the indices. In the revised version of the manuscript, we will include the new result analysis and conclusions.

(b)      In the revised Fig. 9 (see below), SPI3 during each crop critical growing months is contrasted with detrended crop yields. SPI3 is continuous throughout the period of time.

(c)      We agree, the table with the correlations is a more straightforward way of discussing the relation. As noted in b), we also include the SPI3 contrasted with detrended crop yields in the revised Fig. 9. SPI3 has been selected because it is the index that presents the highest correlation with crop yields (see Table 3 below).

(d)      Figure 9 has been redone with additional details. As noted in the first paragraph, while climate conditions are important factors affecting crop yields, they are not the only ones. The relations between SPI-SSI and crop yields are better analyzed now with the new Table 3 and the revised version of Fig 9.

All these clarifications, together with the revised Fig. 9, will be incorporated in the revised manuscript.

| Province | Crop | Indices | | | |
|---|---|---|---|---|---|
| | | SPI3 | SPI6 | SSI3 | SSI6 |
| | wheat | 0.15 | 0.05 | 0.17 | 0.15 |
| Santa Fe | corn | 0.67 | 0.58 | 0.58 | 0.40 |
| | soybean | 0.68 | 0.58 | 0.62 | 0.52 |

| Province | Crop | Indices | | | |
|---|---|---|---|---|---|
| | | SPI3 | SPI6 | SSI3 | SSI6 |
| | wheat | 0.49 | 0.04 | 0.44 | 0.38 |
| Córdoba | corn | 0.60 | 0.55 | 0.51 | 0.55 |
| | soybean | 0.73 | 0.70 | 0.58 | 0.70 |

**Table 3**: Correlation Coefficients of the annual detrended crop yield and the maximum or minimum index value for critical crop months (ON for wheat, DJ for corn and JF for soybean). Maximum or minimum index values are identified according to whether detrended annual crop yields are negative or positive.

[Figure]

**Figure 2:** Percentage of months under moderate to extreme drought conditions (months below one standard deviation) of the total months from January 1979 to December 2018, according to: (a) SPI1, (b) SPI3, (c) SPI6 and (d) soil moisture anomalies.

[Figure]

**Figure 3**: Percentage of months under moderate to extreme drought conditions (months below one standard deviation) accounted during each crop critical growing months from January 1979 to December 2018. For SPI3: (a) wheat during (Oct-Nov), (b) corn during (Dec-Jan), (c) soybean during (Jan-Feb). Panels (d), (e), and (f) showing the same for SPI6.

[Figure]

**Figure 9:** Panels a-c, show the time series of the area-averaged annual crop yield over the provinces of Santa Fe (blue lines) and Córdoba (orange lines) from 1979 to 2018 for: (a) wheat, (b) corn, and (c) soybean. Cubic polynomial trends of each province crop yield time series are in dot line. Panels d-f, presents the detrended time series together with SPI3 index values during each crop critical growing months (ON for wheat, DJ for corn, JF for soybean).

**References.**

Changnon, S. A. (1987). Detecting drought conditions in Illinois. ISWS/CIR-169/87, *Circular no. 169*. http://hdl.handle.net/2142/94485.

Entekhabi, D., Rodriguez-Iturbe, I., & Castelli, F. (1996). Mutual interaction of soil moisture state and atmospheric processes. *Journal of Hydrology*, *184*(1-2), 3-17. https://doi.org/10.1016/0022-1694(95)02965-6.

McNab, A. L. (1989). Climate and drought. *Eos, Transactions American Geophysical Union*, *70*(40), 873-883. https://doi.org/10.1029/89EO00305.

Van Loon, A. F. (2015). Hydrological drought explained. *Wiley Interdisciplinary Reviews: Water*, *2*(4), 359-392. https://doi.org/10.1002/wat2.1085.

---

## Author Comment (AC2) · 19 Dec 2020

The author would like to thank the reviewer for the constructive and thoughtful comments and suggestions which led to substantial improvements in the revised version ofthe manuscript.

Attached please find our response.

Sincerely, Leandro Sgroi

Please also note the supplement to this comment:
https://hess.copernicus.org/preprints/hess-2020-236/hess-2020-236-AC2-supplement.pdf

[Figure]

[Figure]

**Supplement:**

**Responses to Reviewer 2**

The author would like to thank the reviewer for constructive and thoughtful comments and suggestions, which led to substantial improvements in the manuscript's revised version.

In the following, the reviewer's comments are addressed point-by-point. The reviewer's comments are marked in bold and italic. The responses follow in the indented text. Modified figures and a new table are presented at the end of the document.

The next version of the manuscript will include all the issues brought up by the reviewer.

**General comments.**

1. While I understand what the authors are trying to do, as it stands now, the manuscript appears to be a collection of statistical analyses related in some way to drought issues, but no clear objective or story is followed. In particular, what is the main goal of the paper? Is it aimed at developing a monitoring system, to provide information on the link between the indicators and crop yield or just provide a climatology? It seems that the research is focused on the main 3 crops but at the end only a qualitative comparison with yields is shown. I recommend to focus and tailor the discussion in one aspect, in particular, the link with crop yields (see some comments below).

This research aims to advance the understanding and impacts of dry episodes on wheat, corn, and soybean yields over the core crop region of Argentina. Each type of crop has its own phenology with different critical periods when drought may significantly impact production. For this reason, it is essential to consider not only seasonal droughts but also those that center on the critical months. The drought climatology based on different indices is essential to identify drought features that the analysis of a single index might miss. In the current study, our drought documentation focuses on their frequency, duration, and severity, and assess their impacts on the yields of the main crops.

The goal of the research has been rewritten accordingly, following a logical storyline that flows better. The reviewer's comments 3 and 4 have helped to this end.

As for the analysis of yields, we have improved the discussion of the link between drought indices and crop yields. In particular, we have added a new Table 3 (below) that quantifies this relationship using correlations between different drought indices in crop critical periods and crop yields. We have also improved Figure 9. Both Table 3 and revised Fig. 9 give quantitative information to analyze drought's potential impacts on crop yields.

The development of a monitoring system is not part of this study. However, this article could be the first step towards that goal.

2. Abstract: The abstract could be improved, it should be rewritten focusing on the main goals of the manuscript and trying to avoid general statements like: "It is of interest to assess the relationship between those properties and the crop yields." Or "As relevant as the drought duration is its timing and severity".

The abstract has been rewritten according to the reviewer's suggestion. It now states the main goal of the paper. General statements have been removed.

**3.** Introduction: In the last two paragraphs of the intro it would be better to focus on the goals of this paper and how this advances the knowledge in this specific subject. Details on drought indicators should be explained in the methods section.**

Thanks for making this point. The last two paragraphs of the introduction section are rewritten, focusing on the goals of the paper. Furthermore, in the introduction's final paragraph, we now clarify the method to address the proposed objectives. The discussion of drought indicators was moved to the method section in the revised manuscript.

**4. Methods section: the description of yield data is somehow misplaced and cuts the logical flow of the description of the meteorological variables. Consider placing it at the end of the section after the description of the standardized indicators.**

Thanks. We have moved the description of yield data to the end of the section.

5. I'm curious to know why this methodology was selected (eqs. 1 and 2). Does this approach perform better than a parametric approach? I guess that the length of the records (40 years) shouldn't be a problem to fit a parametric distribution. Please motivate this decision and provide a more critical discussion on the methodologies adopted, including advantages and possible shortcomings.

Correct. The 40-year time series of monthly precipitation or soil moisture data are long enough to fit a parametric distribution. A Kolgomorov-Smirnov test reveals that a sample of size 480 (40 years of monthly data) will stay within 6% of the actual distribution with 95% confidence (Massey 1951).

Much of the literature that employs SPI estimation tends to use a 2-parameter Gamma function. Other authors use a 3-parameters Pearson function. In the parametric approach, some theoretical functions do not cope well when precipitation is 0. Further, it has been observed that parametric methods may overestimate or underestimate the extremes. A growing body of research attests that for studies of droughts, a non-parametric approach is better than the parametric one. Unlike parametric approaches, non-parametric methods do not rely on any theoretical distribution. Accordingly, non-parametric methods are valid in a broader range of situations with fewer validity conditions (Siegel, 1957).

Parametric and non-parametric (empirical) probability density functions tend to have differences in the tails, where the parametric distribution may not be a good fit (Farahmand and AghaKouchak 2015). A comparison of parametric and nonparametric estimates of SPI (Soláková et al. 2014) found that differences can be significant in terms of drought severity and not as much in terms of duration. According to Mallenahalli (2020), the non-parametric SPI can better categorize the drought classes, representing better the extent of dryness and normality conditions than parametric approaches.

For these reasons, we adopted a non-parametric methodology that uses an empirical function (Gringorten 1963, Farahmand and AghaKouchak 2015). This method circumvents the use of theoretical functions, avoids issues with 0 precipitation values not uncommon in the study region, and suits well in precipitation and soil moisture studies. Lastly, it is also an opportunity to provide a different approach in the index construction that has not been tested yet in the region.

This discussion is summarized in the revised version of the manuscript.

6. Section 2.3: This is a key section that needs to be strengthened. The drought definition (in the first paragraph) is quite general and vague and it is not clear how this is implemented in the following description. Please be explicit here, e.g. "in our framework a drought starts when our drought indicators are below 1 sd and last until it reaches (threshold) again... ". Moreover, it seems that only one month with negative values of the indicators defines a drought. This is not the most accurate definition of droughts, which need some time to become established, in particular with the threshold chosen (-0.5). However, there are some cases where droughts can develop quickly, e.g. flash droughts and lead to large impacts, but not sure this is reflected here. Please, again, motivate clearly your drought definition. This is key to follow the results.

Section 2.3 has been rewritten to clarify the definition of drought adopted in this study. The revised manuscript states more visibly (in the second paragraph of Section 2.3):

"In this study, droughts are defined as those periods when SPI or SSI depart from the mean at least by minus one Standard Deviation (SD). Droughts last as long as they continue to exceed the threshold. Drought events detected below one standard deviation range from moderate to extreme droughts (McKee et al., 1995). Weaker or milder droughts were estimated using a threshold of 0.5 SD. We also examined different periods, starting at one month and longer. We included the one-month results in Fig. 7 for completeness, but most of our analysis and conclusions are based on longer periods."

To complement the drought definition, we added (in the first paragraph of section 2.3).

"A drought is a sustained period of below-normal water availability (Tallaksen and Van Lanen, 2004; Van Loon, 2015). Droughts are defined as meteorological when there is a precipitation deficit over a period of time. A continued precipitation deficit leads to scarcity of soil moisture that does not meet the plants' water demand. In this case, the drought is called agricultural. This study focuses on meteorological and agricultural droughts and their impacts on crop yields within the region of interest."

Flash droughts are an attractive topic that would require higher frequency data (e.g., daily). As such, it falls outside the focus of this study.

7. The use of precipitation anomalies is not the best approach to make a regional assessment. How do you compute those? I guess P-P\*. This procedure could give misleading results, as the anomalies are not comparable between regions or even different months/seasons. Let's suppose you have one locality with observed 50 mm in one month where the monthly mean is 100 mm, then the anomaly is -50 mm, then for another locality you have observed 5 mm but the monthly mean is 55mm, here your anomalies are again -50 mm. How these deficits are comparable? How do you include those in the areal average? If the goal is to capture the high frequency variations a more suitable indicator would be the percent of Normal precipitation or the SPI-1. Please, for further discussion on that matter refer to the WMO-GWP handbook of drought indicators and indices and references in there.

We appreciate this suggestion. Indeed, precipitation anomalies had been calculated as indicated by the reviewer: P-P\*. As suggested, we replaced the precipitation anomalies by SPI1, and Figs. 2 and 4 were redone (see below). Modifications in Figure 4 also lead to a revised Table 1.

Now, the first paragraph on section 3.1.1 states:

"Figure 2 presents the spatial distribution of drought's occurrence percentage for northern Argentina as characterized by SPI1, SPI3, SPI6, and soil moisture anomalies lower than one time their corresponding standard deviations. The occurrence of drought in northeastern Argentina ranges between 12 and 14% for SPI1 (Fig. 2a), while drought percentages seem to increase up to 18% as characterized by SPI3/6 in the same region (Figs. 2b and 2c). Soil moisture anomalies show that droughts are distributed mainly in the north of Argentina, with percentage values ranging between 16% and 18%. Droughts, as characterized by SPI1/3/6 (Figs. 2a-c), reveal a homogeneous spatial distribution and increasing percentages as with the time scale of the indicator. In contrast, the spatial pattern of soil moisture anomalies shows a decrease in drought percentages for arid regions (Fig. 2d). Inside the Core Crop Region, droughts are more frequent towards the north, with percentages from 14% to 16% for SPI1/3 (see Figs. 2a-b). Fig. 2d indicates that the percentage of droughts, as represented by SPI6, is 18% towards the region's north and southwest. Conversely, drought presence declines towards southeastern CCR as all SPI, and SM show percentages descend to 12% (Figs. 2a-d)."

The new Figure 4 and Table 1 offer better descriptions than in the previous version. SPI1, SPI3, SPI6, and SM anomalies present a common decadal mode that modulates the short frequency interannual variability. These results enable us to improve the discussion in Section 3.1.2 in the revised manuscript.

Section 3.1.2 now states:

"Figure 4 presents the time series SPI1, SPI3, SPI6 as well as soil moisture anomalies, area-averaged over the Core Crop Region. SPI indices and soil moisture (Figs 4a-d) correctly identify wet and dry periods and their interannual variability. Soils function as a physical filter because the output signal (soil moisture) has a lower frequency variability than the input precipitation. Notably, as the SPI time scale increases (from 1 month to 6 months), the variability is reduced (see Figs. 4a and 4c). Table 1 summarizes the dominant modes of interannual variability for SPI1, SPI3, SPI6, and soil moisture. They are (i) a trend, (ii) a band with decadal periodicities, and (iii) another band close to 2.3 years periodicities. Trends explain different percentages of the total variability of the series. Interannual modes in both bands can explain 35% of the total variability of the SPI6 series and 37% of the soil moisture variability.

Decadal cycles in SPI and soil moisture series are closely related and reflect the dry periods of 1987-1991, 1994-1999, and 2004-2013 (see Figs. 4a-d). The short-term 2.3-year cycle of interannual variability is evidenced by frequent wet and dry events between 2000 and 2018 (see Fig. 4b-d). Interestingly, higher amplitudes are noticed starting around 2000. This result agrees with Lovino et al. (2018a, b), who suggested that short-term variability (2.5- to 4-year periods) in precipitation exhibits a large increase in amplitude after 2000.

8. Section 3.1.1. There is an issue with the results in this section, probably due to the lack of drought definition (particularly Figs 2 and 3). The authors compute the probability of occurrence over standardized indicators which means that the probability to be below -1 sd should be 0.1587 if these indicators follow a perfect normal distribution.

I'm afraid that this is what is shown here with the spatial variability due to some statistical instability.

Here a more meaningful series of maps could be the return period of droughts or even the number of drought events per pixel.

The definition of drought was not prominently stated in the manuscript. We have corrected that. As also noted in Comment 6, now, section 2.3 states:

"In this study, droughts are defined as those periods when SPI or SSI depart from the mean at least by minus one standard deviation (SD). Droughts last as long as they continue to exceed the threshold. Drought events detected below one standard deviation range from moderate to extreme droughts (McKee et al., 1995). Weaker or milder droughts were estimated using a threshold of 0.5 SD."

First, apologies, there was an incorrect description. Figures 2 and 3 are not probabilities. They are the number of dry months as a percentage of the total months. This is done for all grid points. Indices are normally distributed, but we did not look at probability.

9. Section 3.1.3. Same issue here as in section 3.1.1. Maybe I'm missing something important here, but what is the added value of showing a histogram for a set of standardized variables?

Even if these represent a regional average, each single point should follow a normal distribution with mean=0 and sd=1, any deviations from this means some problems with the fitting or the input variables (marked dry season, many zeros, etc.). Please clarify, adapt your description towards potential problems in the fitting or add additional relevant information.

In hindsight, there was no need to show histograms of standardized variables, and they have been removed from Fig. 5. The relevant information can be obtained from the P and SM histograms.

10. Section 3.2: There are many studies that relates climate variables and droughts to losses on crop yields in the region (Holzman et al 2014; Podesta et al. 2009; Scian and Bouza 2005; Seiler et al 2007 to cite but a few). This section presents a qualitative description of negative impacts of droughts on crops. Please, clarify how these results advance the literature on the topic in the region or how they are relevant in the context of this study. I believe that the data presented are valuable and could help to answer some questions proposed here, such as what indicator is most appropriate to predict crop losses. For example, a correlation analysis/ contingency table, etc. between the different indicators (and specific dates sensitive to the crops) and the crop losses would make this section much more relevant. This could support some affirmations like the ones on lines 396-402. At the moment the highly sensitive months are presented a-priori.

The novelty is in our severity analysis, and our findings as now quantified with correlations and other supporting material. We present a new Table 3 (see near the end of this document) with the correlation coefficients between detrended crop yields and SPI/SSI during crops critical periods. Table 3 allows quantifying the link between crop yield reductions and drought indicators and giving better support to our findings. Table 3 shows that the shorter-scale indicators (SPI3 and SSI3) achieve a better correlation with crop yields than the larger-scale indicators (SPI6 and SSI6). Then, SPI3 and SSI3 result most appropriate to predict crop yield losses. The relation between detrended crop yields and SPI3 is now presented in the revised Figure 9 (see below).

The results suggest a sensitive relation between summer crops (corn and soybean) and deficits in precipitation and soil moisture during both crops' critical periods. Of the three crops, wheat yields have the lowest correlations with the indices. In the revised version of the manuscript, we will include the new result analysis and conclusions and also discuss our results in the context of previous studies that relate climate variables and droughts to crop losses in the region.

11. The affirmation that corn crop is the most sensitive to water deficits is due to the fact that it is the crop with higher yield/ha (more than 2x to than the other according to lines 323-324). Relative losses related to their average yield values would make their values comparable or even could be greater for the other two crops. As you mention lately corn is also highly sensitive to heat waves as well. Compound events could intensify these losses, but this is not discussed here.

Our text was not clear enough, not precisely in 323-324 that is a description of Fig. 9a-c, but in 355-360 where corn was singled out as being more sensitive

than the other two crops. We have replaced that text with a more accurate and straightforward statement, supported by the correlations in Table 3. The new text replacing 355-360 states:

The detrended time series (Fig. 9d-f) show declines in production due to major drought events. The losses in production may reach up to 1500 kg ha-1 for corn and between 500 to 1000 kg ha-1 for wheat and soybean. Correlations between SPI3 and the different crop yields (Table 3) suggest that corn and soybean are more sensitive to water availability.

Our analysis did not look into compound effects (water scarcity and heat waves) that have a higher impact. As there is earlier literature discussing this topic, we added a brief discussion on compound events based on a previous study by Llano and Vargas (2016a). They show that in central-eastern Argentina, the compound event of precipitation and maximum temperature in corn sensitive growing period have the greatest influences on crop production.

This discussion will be added to the revised manuscript.

**Particular comments.**

12. Line 17: "However, if a multiyear drought experienced breaks, each period would be considered a separate case" this is not clear, please consider rephrasing or even removing this from the abstract.

Thanks, the statement has been removed.

13. Lines 19-20: "Even short dry spells may have large impacts if they occur at the time of the critical growth period of a given crop" this is not discussed or demonstrated in the manuscript. Please, if this is a general statement, consider removing it from the abstract or perform more analysis that can support this affirmation.

The statement has been removed. The abstract will be rewritten in the revised version.

14. Lines 21-22: This also seems a general and rather speculative statement. Consider removing it from this section or perform more analysis that can support this affirmation (see my comment for section 3.2).

The statement has been rephrased to: "*It is shown that severity during the cropsensitive months is a useful indicator for planning purposes.*"

15. Lines 122-126: The performance of a model in the representation of soil moisture is not trivial. Here, there is no mention on how the GLDAS performs in the region. A formal validation is not necessary but at least some references describing its validity in the region are needed. See for instance, Spennemann et al., 2015 & 2020.

We now reference studies that evaluate GLDAS performance in the area. Spennemann et al. (2015; 2020) reported that GLDAS reproduces the observed soil moisture patterns satisfactorily. They also found that land surface model derived indices using GLDAS can be used as soil monitoring indices in agricultural production management. Grings et al. (2015) evaluated satellitederived estimations against in situ observations, suggesting that GLDAS seems to be a good soil moisture benchmark in the Pampas region.

The text in the revised manuscript will state:

"The monthly precipitation data from January 1979 to December 2018 employed here was developed by NCEP's Climate Prediction Center (CPC) and consists of in situ observations spatially interpolated to a regular 0.5°× 0.5° latitude-longitude grid cell (Chen et al., 2008). This product has been used as a benchmark for the precipitation model's comparison in South America (Silva et al., 2011).

In the absence of soil moisture observations, we employ products obtained from the Global Land Data Assimilation System (GLDAS; Rodell et al., 2004; Meng et al., 2012; Beaudoing and Rodell, 2019; 2020). In GLDAS, soil moisture is derived from the surface water and energy balances forced by observations in a land surface model (Noah in this case). The Noah Model considers four soil layers (0-10 cm, 10-40 cm, 40-100 cm, and 100-200 cm) totaling 2 meters depth for which the SM total column value was determined (Rodell et al., 2004). The data set consists of soil moisture monthly values at a spatial resolution of 0.25°× 0.25° over the same period of analysis. Recent studies (see, e.g., Grings et al., 2015, and Spennemann et al., 2015; 2020) evaluated GLDAS soil moisture products in the Humid Pampas with good results. According to Grings et al. (2015), GLDAS seems to be a good soil moisture benchmark in the Pampas region. Also, Spennemann et al. (2015, 2020) reported that GLDAS reproduces well soil moisture observational patterns. They also found that GLDAS land surface model derived indices can be used as soil monitoring indices in the context of agricultural production management".

**16. Line 47: This only apply to dry events; wet events has a completely different dynamic. Please, clarify.**

We have removed the reference to wet periods from the sentence. It now states: "Statistical analysis of extreme events in SESA has shown that periods of water deficit occur at different time scales, with an inverse relationship between frequency and duration, i.e., more frequent events tend to be shorter-lived."

**17. Line 69: Agriculture will be only one sector affected. Droughts can affect almost any compartment of economy and ecosystems.**

We have simplified the sentence to: "If confirmed, such change could lead to more frequent droughts."

**18. Line 104: the sentence between [: : :] should be removed of placed elsewhere.**

Thank you. The sentence has been removed.

**19. Line 203-205: how the soil moisture can be affected by soil degradation and desertification? Is this modelled in GLDAS?**

We have removed the speculative sentence and their references from the manuscript. GLDAS does not consider changes in soils and their properties, and therefore cannot address soil degradation issues. Recent studies have started looking into desertification with a combination of datasets that include GLDAS (e.g., Fan et al. 2020).

20. Line 207: The SPI-6 could be correlated with hydrological droughts (probably longer aggregation periods can do it better) but it is still a meteorological indicator as relies only on precipitation. It is not accurate to attribute the representation of hydrological droughts to it. Please consider rephrasing this sentence.

We agree with the reviewer. More extended periods like in SPI-6 can reflect better a hydrological drought than the shorter periods of SPI-3, but it indeed relies only on precipitation. The sentence removed the reference to hydrological drought.

"Fig. 2d indicates that the percentage of droughts, as represented by SPI6, is 18% towards the region's north and southwest."

**21. Line 216-217: How the maps in Fig 3 can be interpreted as the temporal evolution of drought frequency?**

The sentence was removed. While the information was in the caption, the identification of the critical periods was not evident from the figure. We have added insets clarifying what part of the year each critical period corresponds (see Fig 3. below).

Wheat is sensitive to drought frequency in late spring (Oct-Nov), corn is sensitive to droughts during summer (Dec-Jan), and soybean is most sensitive during Jan-Feb. Therefore, it does not describe a temporal evolution of drought frequency, as our sentence claimed. It is the shift in relevant drought months for each crop type.

22. Lines 314-315: "Therefore, droughts in the Core Crop Region are detected more easily when using SSI instead of SPI." It is not possible to benchmark drought detection by just comparing two drought indicators. Just because one indicator covers more area or is more severe, etc., doesn't mean that is the best performing indicator. Instead, I would argue that the best indicator should be the one that better represents the specific sectorial impacts, in this case, crop production.

Good point. We have rephrased it: "Droughts affecting crop growth are detected either with SPI or SSI indices. This is the case of drought events occurring in 1988, 1997, 2007, 2009, and 2017 that spread over 80% to 90% of the region (Fig. 8d)."

**References.**

Beaudoing, H.,and Rodell, M. (2019). GLDAS Noah Land Surface Model L4 monthly 0.25 x 0.25 degree V2.0, Greenbelt, Maryland, USA, Goddard Earth Sciences Data and Information Services Center (GES DISC), NASA/GSFC/HSL, Accessed: [12 March 2019], https://doi.org/10.5067/9SQ1B3ZXP2C5.

Beaudoing, H., and Rodell, M. (2020). GLDAS Noah Land Surface Model L4 monthly 0.25 x 0.25 degree V2.1, Greenbelt, Maryland, USA, Goddard Earth Sciences Data and Information Services Center (GES DISC), NASA/GSFC/HSL, Accessed: [12 March 2019], https://doi.org/10.5067/SXAVCZFAQLNO.

Chen, M., Shi, W., Xie, P., Silva, V. B., Kousky, V. E., Wayne Higgins, R., & Janowiak, J. E. (2008). Assessing objective techniques for gauge-based analyses of global daily precipitation. Journal of Geophysical Research: Atmospheres, 113(D4).

Fan, Z.; Li, S. (2020). Fang, H. Explicitly Identifying the Desertification Change in CMREC Area Based on Multisource Remote Data. Remote Sens. 2020, 12, 3170. https://doi.org/10.3390/rs12193170.

Farahmand, A., and AghaKouchak, A. (2015). A generalized framework for deriving nonparametric standardized drought indicators. Adv. Water Resour, 76, 140-145, https://doi.org/10.1016/j.advwatres.2014.11.012.

Gringorten, I. I. (1963). A plotting rule for extreme probability paper. J. Geophys. Res., 68, 813-814, https://doi.org/10.1029/JZ068i003p00813.

Grings, F., Bruscantini, C. A., Smucler, E., Carballo, F., Dillon, M. E., Collini, E. A., ... & Karszenbaum, H. (2015). Validation strategies for satellite-based soil moisture products over Argentine Pampas. IEEE Journal of Selected Topics in Applied Earth Observations and Remote Sensing, 8(8), 4094-4105.

Holzman, M. E., Rivas, R., & Piccolo, M. C. (2014). Estimating soil moisture and the relationship with crop yield using surface temperature and vegetation index. International Journal of Applied Earth Observation and Geoinformation, 28, 181-192.

Llano, M.P. and Vargas, W. (2016), Climate characteristics and their relationship with soybean and maize yields in Argentina, Brazil and the United States. Int. J. Climatol., 36: 1471-1483. https://doi.org/10.1002/joc.4439.

Mallenahalli, N.K. (2020). Comparison of parametric and non-parametric standardized precipitation index for detecting meteorological drought over the Indian region. Theor Appl Climatol 142, 219–236. https://doi.org/10.1007/s00704-020-03296-z.

Massey Jr, F. J. (1951). The Kolmogorov-Smirnov test for goodness of fit. Journal of the American statistical Association, 46(253), 68-78.

Meng, J., Yang, R., Wei, H., Ek, M., Gayno, G., Xie, P., & Mitchell, K. (2012). The land surface analysis in the NCEP Climate Forecast System Reanalysis. Journal of Hydrometeorology, 13(5), 1621-1630.

Podestá, G., Bert, F., Rajagopalan, B., Apipattanavis, S., Laciana, C., Weber, E., ... & Menendez, A. (2009). Decadal climate variability in the Argentine Pampas: regional impacts of plausible climate scenarios on agricultural systems. Climate Research, 40(2-3), 199-210.

Rodell, M., Houser, P. R., Jambor, U. E. A., Gottschalck, J., Mitchell, K., Meng, C. J., ... & Entin, J. K. (2004). The global land data assimilation system. Bulletin of the American Meteorological Society, 85(3), 381-394.

Scian, B. V., & Bouza, M. E. (2005). Environmental variables related to wheat yields in the semiarid pampa region of Argentina. Journal of arid environments, 61(4), 669-679.

Seiler, R. A., Kogan, F., Wei, G., & Vinocur, M. (2007). Seasonal and interannual responses of the vegetation and production of crops in Cordoba–Argentina assessed by AVHRR derived vegetation indices. Advances in Space Research, 39(1), 88-94.

Silva, V. B., Kousky, V. E., & Higgins, R. W. (2011). Daily precipitation statistics for South America: An intercomparison between NCEP reanalyses and observations. Journal of Hydrometeorology, 12(1), 101-117.

Soláková T, De Michele C, Vezzoli R (2014). Comparison between parametric and nonparametric approaches for the calculation of two drought indices: SPI and SSI. J Hydrol Eng 19(9):04014010. https://doi.org/10.1061/(ASCE)HE.1943-5584.0000942.

Spennemann, P. C., Rivera, J. A., Saulo, A. C., & Penalba, O. C. (2015). A comparison of GLDAS soil moisture anomalies against standardized precipitation index and multisatellite estimations over South America. Journal of Hydrometeorology, 16(1), 158-171.

Spennemann, P. C., Fernández-Long, M. E., Gattinoni, N. N., Cammalleri, C., & Naumann, G. (2020). Soil moisture evaluation over the Argentine Pampas using models, satellite estimations and in-situ measurements. Journal of Hydrology: Regional Studies, 31, 100723.

Tallaksen, L. M., & Van Lanen, H. A. (Eds.). (2004). Hydrological drought: processes and estimation methods for streamflow and groundwater (Vol. 48). Elsevier.

Van Loon, A. F. (2015). Hydrological drought explained. Wiley Interdisciplinary Reviews: Water, 2(4), 359-392. https://doi.org/10.1002/wat2.1085

WMO, G.; GWP, G. Handbook of Drought Indicators and Indices. Geneva: World Meteorological Organization (WMO) and Global Water Partnership (GWP), 2016.

|                         | SPI1 | SPI3 | SPI6 | SM
Anomalies |
|-------------------------|------|------|------|-----------------|
| Trend                   | -    | 3.2  | 5.5  | -               |
| Quasi-cycle, T ~ 10 yr  | 5.9  | 13.9 | 22   | 25.4            |
| Quasi-cycle, T ~ 2.3 yr | -    | 7.8  | 13   | 11.6            |

Table 1: Percentage of variance explained by the dominant modes of interannual variability detected using SSA with a window length of 120 months. Series of SPI1, SPI3, SPI6 and soil moisture anomalies from January 1979 to December 2018 in Core Crop Region.

| Province | Crop -  | Indices |      |      |      |  |
|----------|---------|---------|------|------|------|--|
|          |         | SPI3    | SPI6 | SSI3 | SSI6 |  |
| Santa Fe | wheat   | 0.15    | 0.05 | 0.17 | 0.15 |  |
|          | corn    | 0.67    | 0.58 | 0.58 | 0.40 |  |
|          | soybean | 0.68    | 0.58 | 0.62 | 0.52 |  |
|          |         |         |      |      |      |  |
| Province | Crop -  | Indices |      |      |      |  |
|          |         | SPI3    | SPI6 | SSI3 | SSI6 |  |
| Córdoba  | wheat   | 0.49    | 0.04 | 0.44 | 0.38 |  |
|          | corn    | 0.60    | 0.55 | 0.51 | 0.55 |  |
|          | soybean | 0.73    | 0.70 | 0.58 | 0.70 |  |

**Table 3**: Correlation Coefficients of the annual detrended crop yield and the maximum or minimum index value for critical crop months (ON for wheat, DJ for corn and JF for soybean). Maximum or minimum index values are identified according to whether detrended annual crop yields are negative or positive.

---

## Author Response (AR1)

**Responses to Reviewers**

The authors would like to thank the reviewers for their constructive and thoughtful comments and suggestions, which led to substantial improvements in the manuscript's revised version.

In the following, the reviewer's comments are addressed point-by-point. The reviewer's comments are marked in bold and italic. The responses follow in the indented text. Modified figures and a new table are presented at the end of the document. The changes are referenced to the lines of the manuscript without marked changes.

**Reviewer 1**

**Some specific aspects about your work:**

***1. In equation 1 and 2, when describing "xi", it would be better if you consider the "i" as a subscript.***

> Corrected in the revised manuscript (see lines 151 and 156).

***2. I think that there is an error in line 205: "that is higher than for precipitation" should be "that is higher for precipitation".***

> We realize that the sentence was confusing. In the revised version of the manuscript, the variable "precipitation anomalies" has been replaced by SPI1.
>
> The entire paragraph has been rewritten. It now states (lines 208 to 218):
>
> *" Figure 2 presents the spatial distribution of the percentage of months under moderate to extreme drought conditions for northern Argentina as characterized by SPI1, SPI3, SPI6, and soil moisture anomalies. The occurrence of drought in northeastern Argentina ranges between 12 and 14% for SPI1 (Fig. 2a), while months with droughts seem to increase up to 18% as characterized by SPI3/6 (Figs. 2b and 2c). Soil moisture anomalies show that droughts are distributed mainly in the north of Argentina, with about 16% - 18% of months with drought. Droughts, as characterized by SPI1/3/6 (Figs. 2a-c), reveal a homogeneous spatial distribution and an increasing drought percentage as with the time scale of the indicator. In contrast, the spatial pattern of soil moisture anomalies shows a decrease in drought percentages for arid regions (Fig. 2d). Inside the Core Crop Region, droughts are more frequent towards the north, with percentages of months under moderate to extreme drought conditions from 14% to 16% for SPI1/3 (see Figs. 2a-b). Fig. 2d indicates that months with drought conditions, as represented by SPI6, are equivalent to 18% towards the region's north and southwest. Conversely, drought presence declines towards the southeastern core crop region as all SPI and SM show percentages descend to 12% (Figs. 2a-d)."*

***3. In Figure 3 the color bar is not clear enough. The top value (0.18) is reached in an important fraction of the area, and the impression is that higher values can also be possible. Maybe changing the color bar and allowing a higher limit can improve that figure. Some of your comments are related to the 0.18 probability, but that can change if a different color bar is used.***

We realized that probability is not the proper expression for Figures 2 and 3. Instead, we now use percentages of drought occurrence. Figures 2 and 3 have been redone accordingly (see figures at the end of the responses).

The drought's occurrence percentage does not exceed 20% in any of the cases. The largest values in Figure 3 are in the range 18 % to 20%. As an example, we present below the class distribution histogram for SPI6 wheat critical months (Fig. 3d). The rest of the maps in Figure 3 present very similar histograms.

To clarify this issue, we have adjusted the color bars in Figs. 2 and 3 by adding the upper maximum limit of 20%.

[Figure]

**4. When you estimate SPI3 and SPI6 for the critical months and analyze the temporal evolution of drought frequency. Is there any overlapping between these months and the estimation of the SPI-SSI values? How does that overlapping affect your conclusions?**

Critical months refer to the specific periods when the crops are most sensitive to ambient conditions. These periods do not necessarily cover the predetermined seasons, e.g., SON or DJF.

Therefore, their analysis is aimed at evaluating the potential of droughts to affect crop yields during those specific periods. Even though there is an overlap with the 3- and 6-months' time scale for indicators, it does not affect any of the conclusions presented in our study.

**5. In _line 230_ you mention that SM acts as a physical filter, but there is no mention on the mechanisms. Maybe exploring the physical processes involved should be incorporated and analyzed.**

In the framework of the surface water budget, soil moisture is estimated as a difference between incoming precipitation and outgoing runoff and evapotranspiration (these are the main terms). Different mechanisms govern the behavior of each of the components as well as how they interact in a temporal scale. When precipitation enters the system, its signal will propagate in time. The fundamental concept was discussed by Changnon (1987) and McNab (1989) using the schematic shown on the side. The schematic shows a time delay between variables and a smoothing of the curves.

[Figure]

Part of the precipitation that reaches the ground will evaporate from the surface, another part will turn into surface runoff, and the rest will infiltrate into the ground. Some of the infiltrated water may go back to the atmosphere through vegetation transpiration of water extracted by roots. Another portion may percolate to groundwater, and the rest will be stored as soil moisture (Changnon, 1987; Van Loon, 2015). The time it takes for the precipitated water to infiltrate the soil and through deeper layers has a dampening or smoothing effect that Entekhabi et al. (2006) described as a low-pass filter.

We have added a brief discussion of this process to the revised manuscript in lines 236 to 238. The analysis of evapotranspiration and runoff, while attractive, would require a different approach (based in land surface models) that is out of the scope of our study.

**6. The Cordoba Province is written as Córdoba and Cordoba (with and without accent).**

The proper spelling is with an accent. We have corrected it throughout the manuscript.

**7. In _Figure 8_, the inclusion of a legend will facilitate the understanding of the meaning of each line.**

Thanks for pointing it out. A new legend has been included in the figure (now Figure 7).

**8. _Figure 9_ shows the detrended crop yield and the SPI-SSI indexes and your conclusions say that (a) lower SPI-SSI values implies crop yield losses, but that is not the case for every year. Is there any explanation? (b) Why are only some years**

*shown? (c) Maybe a direct comparison between the SPI-SSI indexes and the crops yields (instead of a time series), considering all the years, can be more descriptive. (d) Your conclusions depend on this figure and should be clearer the relation between SPI-SSi and crop-yields and explain years without reasonable results, like the 2004-2009 period.*

(a)    Climate conditions (as those given by SPI or SSI) affect crop yields, but they are not the only factors at play. Non-climatic factors like seed quality, different use of fertilizers, sawing or harvesting dates are a few of the possible effects on crop yields. This issue has been clarified in the revised manuscript. See lines 315-316.

A better way to support our results is by computing correlations between the drought indices and crop yields. The new Table 3 (see at the end of the responses) shows that indeed there is a relation between indices (SPI3/6 and SSI3/6) and crop yields. Table 3 shows that the shorter-scale indicators (SPI3 and SSI3) achieve a better correlation with crop yields than the larger-scale indicators (SPI6 and SSI6). Then, SPI3 and SSI3 result most appropriate to predict crop yield losses. The relation between detrended crop yields and SPI3 is now presented in the revised Figure 8 (see at the end of the responses).

The results suggest a sensitive relation between summer crops (corn and soybean) and deficits in precipitation and soil moisture during both crops' critical periods. Of the three crops, wheat yields have the lowest correlations with the indices. In the revised version of the manuscript, we include the new result analysis and conclusions (see lines 326-332; 396-397 and 402-406).

(b)    In the revised Fig. 8 (see at the end of the responses), SPI3 during each crop critical growing months is contrasted with detrended crop yields. SPI3 is continuous throughout the period of time.

(c)    We agree, the table with the correlations is a more straightforward way of discussing the relation. As noted in b), we also include the SPI3 contrasted with detrended crop yields in the revised Fig. 8. SPI3 has been selected because it is the index that presents the highest correlation with crop yields (see Table 3 at the end of the responses).

(d)    Figure 9 (now Figure 8) has been redone with additional details. As noted in the first paragraph, while climate conditions are important factors affecting crop yields, they are not the only ones. The relations between SPI-SSI and crop yields are better analyzed now with the new Table 3 and the revised version of Fig 8.

**References**

Changnon, S. A. (1987). Detecting drought conditions in Illinois. ISWS/CIR-169/87, *Circular no. 169.* http://hdl.handle.net/2142/94485.

Entekhabi, D., Rodriguez-Iturbe, I., & Castelli, F. (1996). Mutual interaction of soil moisture state and atmospheric processes. *Journal of Hydrology*, *184*(1-2), 3-17. https://doi.org/10.1016/0022-1694(95)02965-6.

McNab, A. L. (1989). Climate and drought. *Eos, Transactions American Geophysical Union*, *70*(40), 873-883. https://doi.org/10.1029/89EO00305.

Van Loon, A. F. (2015). Hydrological drought explained. *Wiley Interdisciplinary Reviews: Water*, *2*(4), 359-392. https://doi.org/10.1002/wat2.1085.

**Responses to Reviewer 2**

*General comments.*

*1. While I understand what the authors are trying to do, as it stands now, the manuscript appears to be a collection of statistical analyses related in some way to drought issues, but no clear objective or story is followed. In particular, what is the main goal of the paper? Is it aimed at developing a monitoring system, to provide information on the link between the indicators and crop yield or just provide a climatology? It seems that the research is focused on the main 3 crops but at the end only a qualitative comparison with yields is shown. I recommend to focus and tailor the discussion in one aspect, in particular, the link with crop yields (see some comments below).*

> This research aims to advance the understanding and impacts of dry episodes on wheat, corn, and soybean yields over the core crop region of Argentina. Each type of crop has its own phenology with different critical periods when drought may significantly impact production. For this reason, it is essential to consider not only seasonal droughts but also those that center on the critical months. The drought climatology based on different indices is essential to identify drought features that the analysis of a single index might miss. In the current study, our drought documentation focuses on their frequency, duration, and severity, and assess their impacts on the yields of the main crops.

> The goal of the research has been rewritten accordingly, following a logical storyline that flows better (see lines 84 to 89). The reviewer's comments 3 and 4 have helped to this end.

> As for the analysis of yields, we have improved the discussion of the link between drought indices and crop yields. In particular, we have added a new Table 3 (see at the end of the responses) that quantifies this relationship using correlations between different drought indices in crop critical periods and crop yields. We have also improved Figure 9 (now Figure 8). Both Table 3 and revised Fig. 8 give quantitative information to analyze drought's potential impacts on crop yields.

> The development of a monitoring system is not part of this study. However, this article could be the first step towards that goal.

*2. Abstract: The abstract could be improved, it should be rewritten focusing on the main goals of the manuscript and trying to avoid general statements like: "It is of interest to assess the relationship between those properties and the crop yields." Or "As relevant as the drought duration is its timing and severity".*

> The abstract has been rewritten according to the reviewer's suggestion. It now states the main goal of the paper. General statements have been removed.

*3. Introduction: In the last two paragraphs of the intro it would be better to focus on the goals of this paper and how this advances the knowledge in this specific subject. Details on drought indicators should be explained in the methods section.*

> Thanks for making this point. The last two paragraphs of the introduction section are rewritten, focusing on the goals of the paper. The discussion of drought indicators was moved to the method section in the revised manuscript (lines 126 to 134).

***4. Methods section: the description of yield data is somehow misplaced and cuts the logical flow of the description of the meteorological variables. Consider placing it at the end of the section after the description of the standardized indicators.***

Thanks. We have moved the description of yield data to the end of the section (see lines 167 to 169).

***5. I'm curious to know why this methodology was selected (eqs. 1 and 2). Does this approach perform better than a parametric approach? I guess that the length of the records (40 years) shouldn't be a problem to fit a parametric distribution. Please motivate this decision and provide a more critical discussion on the methodologies adopted, including advantages and possible shortcomings.***

Correct. The 40-year time series of monthly precipitation or soil moisture data are long enough to fit a parametric distribution. A Kolgomorov-Smirnov test reveals that a sample of size 480 (40 years of monthly data) will stay within 6% of the actual distribution with 95% confidence (Massey 1951).

Much of the literature that employs SPI estimation tends to use a 2-parameter Gamma function. Other authors use a 3-parameters Pearson function. In the parametric approach, some theoretical functions do not cope well when precipitation is 0. Further, it has been observed that parametric methods may overestimate or underestimate the extremes. A growing body of research attests that for studies of droughts, a non-parametric approach is better than the parametric one. Unlike parametric approaches, non-parametric methods do not rely on any theoretical distribution. Accordingly, non-parametric methods are valid in a broader range of situations with fewer validity conditions (Siegel, 1957).

Parametric and non-parametric (empirical) probability density functions tend to have differences in the tails, where the parametric distribution may not be a good fit (Farahmand and AghaKouchak 2015). A comparison of parametric and non-parametric estimates of SPI (Soláková et al. 2014) found that differences can be significant in terms of drought severity and not as much in terms of duration. According to Mallenahalli (2020), the non-parametric SPI can better categorize the drought classes, representing better the extent of dryness and normality conditions than parametric approaches.

For these reasons, we adopted a non-parametric methodology that uses an empirical function (Gringorten 1963, Farahmand and AghaKouchak 2015). This method circumvents the use of theoretical functions, avoids issues with 0 precipitation values not uncommon in the study region, and is suitable in precipitation and soil moisture studies. Lastly, it is also an opportunity to provide a different approach in the index construction that has not been tested yet in the region.

This discussion is summarized in the revised version of the manuscript (see lines 136 to 147).

***6. Section 2.3: This is a key section that needs to be strengthened. The drought definition (in the first paragraph) is quite general and vague and it is not clear how this is implemented in the following description. Please be explicit here, e.g. "in***

*our framework a drought starts when our drought indicators are below 1 sd and last until it reaches (threshold) again… ". Moreover, it seems that only one month with negative values of the indicators defines a drought. This is not the most accurate definition of droughts, which need some time to become established, in particular with the threshold chosen (-0.5). However, there are some cases where droughts can develop quickly, e.g. flash droughts and lead to large impacts, but not sure this is reflected here. Please, again, motivate clearly your drought definition. This is key to follow the results.*

Section 2.3 has been rewritten to clarify the definition of drought adopted in this study. The revised manuscript states more visibly (in the second paragraph of Section 2.3, lines 177 to 181):

*" For this analysis, droughts are defined as those periods when SPI or SSI depart from the mean at least by minus one standard deviation. Drought events below that threshold range from moderate to extreme droughts (McKee et al., 1995). Weaker or milder droughts were estimated using a threshold of one half the standard deviation. Droughts persist as long as they continue to exceed the threshold. We also examined different periods, starting at one month and more prolonged. We included the one-month results in Fig. 6 for completeness, but most of our analysis and conclusions are based on longer periods."*

To complement the drought definition, we added (in the first paragraph of section 2.3, lines 171 to 175).

*" A drought is a sustained period of below-normal water availability (Tallaksen and Van Lanen, 2004; Van Loon, 2015). Droughts are identified as "meteorological droughts" when there is a precipitation deficit over a period of time. A continued precipitation deficit can lead to a scarcity of soil moisture that does not meet the plants' water demand. In this case, the drought is called "agricultural drought". This study focuses on meteorological and agricultural droughts and their impacts on crop yields within the region of interest."*

Flash droughts are an attractive topic that would require higher frequency data (e.g., daily). As such, it falls outside the focus of this study.

*7. The use of precipitation anomalies is not the best approach to make a regional assessment. How do you compute those? I guess P-P\*. This procedure could give misleading results, as the anomalies are not comparable between regions or even different months/seasons. Let's suppose you have one locality with observed 50 mm in one month where the monthly mean is 100 mm, then the anomaly is -50 mm, then for another locality you have observed 5 mm but the monthly mean is 55mm, here your anomalies are again -50 mm. How these deficits are comparable? How do you include those in the areal average? If the goal is to capture the high frequency variations a more suitable indicator would be the percent of Normal precipitation or the SPI-1. Please, for further discussion on that matter refer to the WMO-GWP handbook of drought indicators and indices and references in there.*

We appreciate this suggestion. Indeed, precipitation anomalies had been calculated as indicated by the reviewer: P-P\*. As suggested, we replaced the precipitation anomalies by SPI1, and Figs. 2 and 4 were redone (see at the end of the responses). Modifications in Figure 4 also lead to a revised Table 1 (see also at the end of the responses).

Now, the first paragraph on section 3.1.1 (lines 208 to 218) states:

" *Figure 2 presents the spatial distribution of the percentage of months under moderate to extreme drought conditions for northern Argentina as characterized by SPI1, SPI3, SPI6, and soil moisture anomalies. The occurrence of drought in northeastern Argentina ranges between 12 and 14% for SPI1 (Fig. 2a), while months with droughts seem to increase up to 18% as characterized by SPI3/6 (Figs. 2b and 2c). Soil moisture anomalies show that droughts are distributed mainly in the north of Argentina, with about 16% - 18% of months with drought. Droughts, as characterized by SPI1/3/6 (Figs. 2a-c), reveal a homogeneous spatial distribution and an increasing drought percentage as with the time scale of the indicator. In contrast, the spatial pattern of soil moisture anomalies shows a decrease in drought percentages for arid regions (Fig. 2d). Inside the Core Crop Region, droughts are more frequent towards the north, with percentages of months under moderate to extreme drought conditions from 14% to 16% for SPI1/3 (see Figs. 2a-b). Fig. 2d indicates that months with drought conditions, as represented by SPI6, are equivalent to 18% towards the region's north and southwest. Conversely, drought presence declines towards the southeastern core crop region as all SPI and SM show percentages descend to 12% (Figs. 2a-d)."*

The new Figure 4 and Table 1 offer better descriptions than in the previous version. SPI1, SPI3, SPI6, and SM anomalies present a common decadal mode that modulates the short frequency interannual variability. These results enable us to improve the discussion in Section 3.1.2 in the revised manuscript.

Section 3.1.2 now states (lines 233 to 248):

*"Figure 4 presents the time series of SPI1, SPI3, SPI6, and soil moisture anomalies, area-averaged over the Core Crop Region. SPI indices and soil moisture (Figs 4a-d) help identify wet and dry periods and their interannual variability. Notably, as the SPI time scale increases (from one month to six months), the variability is reduced (see Figs. 4a and 4c). Soils function as a physical filter because the output signal (soil moisture) has a lower frequency variability than the input precipitation. The reason is that the time it takes for the precipitated water to infiltrate the soil and move through deeper layers has a dampening or smoothing effect that Entekhabi et al. (2006) described as a low-pass filter.*

*The main features in Fig 4 are summarized in Table 1 that reveals the dominant modes of interannual variability for SPI1, SPI3, SPI6, and soil moisture. They are (i) a trend, (ii) a band with decadal periodicities, and (iii) a band close to 2.3 years periodicities. Trends explain different percentages of the total variability of the series. Interannual modes in both bands can explain 35% of the total variability of the SPI6 series and 37% of the soil moisture variability. Decadal cycles in SPI and soil moisture series are closely related and reflect the dry periods of 1987-1991, 1994-1999, and 2004-2013 (see Figs. 4a-d). The short-term 2.3-year cycle of interannual variability is evidenced by frequent wet and dry events between 2000 and 2018 (see Fig. 4b-d). Interestingly, higher amplitudes are noticed starting around 2000. This result agrees with Lovino et al. (2018a, b), who suggested that short-term variability (2.5- to 4-year periods) in precipitation exhibits a large increase in amplitude after 2000."*

*8. Section 3.1.1. There is an issue with the results in this section, probably due to the lack of drought definition (particularly Figs 2 and 3). The authors compute the probability of occurrence over standardized indicators which means that the probability to be below -1 sd should be 0.1587 if these indicators follow a perfect normal distribution.*
*I'm afraid that this is what is shown here with the spatial variability due to some statistical instability.*
*Here a more meaningful series of maps could be the return period of droughts or even the number of drought events per pixel.*

> The definition of drought was not prominently stated in the manuscript. We have corrected that. As also noted in Comment 6, now, section 2.3 states (lines 177 to 180):

> *"For this analysis, droughts are defined as those periods when SPI or SSI depart from the mean at least by minus one standard deviation. Drought events below that threshold range from moderate to extreme droughts (McKee et al., 1995). Weaker or milder droughts were estimated using a threshold of one half the standard deviation. Droughts persist as long as they continue to exceed the threshold."*

> First, apologies, there was an incorrect description. Figures 2 and 3 are not probabilities. They are the number of dry months as a percentage of the total months. This is done for all grid points. Indices are normally distributed, but we did not look at probability.

*9. Section 3.1.3. Same issue here as in section 3.1.1. Maybe I'm missing something important here, but what is the added value of showing a histogram for a set of standardized variables?*
*Even if these represent a regional average, each single point should follow a normal distribution with mean=0 and sd=1, any deviations from this means some problems with the fitting or the input variables (marked dry season, many zeros, etc.). Please clarify, adapt your description towards potential problems in the fitting or add additional relevant information.*

> In hindsight, there was no need to show histograms of standardized variables, and they have been removed from Fig. 5. The relevant information can be obtained from the P and SM histograms.

*10. Section 3.2: There are many studies that relates climate variables and droughts to losses on crop yields in the region (Holzman et al 2014; Podesta et al. 2009; Scian and Bouza 2005; Seiler et al 2007 to cite but a few). This section presents a qualitative description of negative impacts of droughts on crops. Please, clarify how these results advance the literature on the topic in the region or how they are relevant in the context of this study. I believe that the data presented are valuable and could help to answer some questions proposed here, such as what indicator is most appropriate to predict crop losses. For example, a correlation analysis/ contingency table, etc. between the different indicators (and specific dates sensitive to the crops) and the crop losses would make this section much more relevant. This could support some affirmations like the ones on lines 396-402. At the moment the highly sensitive months are presented a-priori.*

The novelty is in our severity analysis, and our findings as now quantified with correlations and other supporting material. We present a new Table 3 (see near the end of this document) with the correlation coefficients between detrended crop yields and SPI/SSI during crops critical periods. Table 3 allows quantifying the link between crop yield reductions and drought indicators and giving better support to our findings. Table 3 shows that the shorter-scale indicators (SPI3 and SSI3) achieve a better correlation with crop yields than the larger-scale indicators (SPI6 and SSI6). Then, SPI3 and SSI3 result most appropriate to predict crop yield losses. The relation between detrended crop yields and SPI3 is now presented in the revised Figure 8 (see at the end of the responses).

The results suggest a sensitive relation between summer crops (corn and soybean) and deficits in precipitation and soil moisture during both crops' critical periods. Of the three crops, wheat yields have the lowest correlations with the indices. In the revised version of the manuscript, we include the new result analysis and conclusions (see lines 326-332; 396-397 and 402-406) and also discuss our results in the context of previous studies that relate climate variables and droughts to crop losses in the region (lines 401 to 408).

**11. The affirmation that corn crop is the most sensitive to water deficits is due to the fact that it is the crop with higher yield/ha (more than 2x to than the other according to lines 323-324). Relative losses related to their average yield values would make their values comparable or even could be greater for the other two crops. As you mention lately corn is also highly sensitive to heat waves as well. Compound events could intensify these losses, but this is not discussed here.**

Our text was not clear enough, not precisely in 323-324 that is a description of Fig. 8a-c, but in 355-360 where corn was singled out as being more sensitive than the other two crops (lines referring to the original manuscript). We have replaced that text with a more accurate and straightforward statement, supported by the correlations in Table 3. The new text replacing 355-360 states (see lines 344 to 346):

*The detrended time series (Fig. 8d-f) show declines in production due to major drought events. The losses in production may reach up to 1500 kg ha$^{-1}$ for corn and between 500 to 1000 kg ha$^{-1}$ for wheat and soybean. Correlations between SPI3 and the different crop yields (Table 3) suggest that corn and soybean are more sensitive to water availability.*

Our analysis did not look into compound effects (water scarcity and heat waves) that have a higher impact. As there is earlier literature discussing this topic, we added a brief discussion on compound events based on a previous study by Llano and Vargas (2016) (see lines 352 to 355). They show that in central-eastern Argentina, the compound event of precipitation and maximum temperature in corn sensitive growing period have the greatest influences on crop production.

**Particular comments.**

**12. Line 17: "However, if a multiyear drought experienced breaks, each period would be considered a separate case" this is not clear, please consider rephrasing or even removing this from the abstract.**

Thanks, the statement has been removed.

**13. Lines 19-20: "Even short dry spells may have large impacts if they occur at the time of the critical growth period of a given crop" this is not discussed or demonstrated in the manuscript. Please, if this is a general statement, consider removing it from the abstract or perform more analysis that can support this affirmation.**

The statement has been removed. The abstract was rewritten in the revised version.

**14. Lines 21-22: This also seems a general and rather speculative statement. Consider removing it from this section or perform more analysis that can support this affirmation (see my comment for section 3.2).**

The statement has been rephrased to (lines 20-21): " *Large drought severity values during the crop-sensitive months result in significant crop yield losses* "

**15. Lines 122-126: The performance of a model in the representation of soil moisture is not trivial. Here, there is no mention on how the GLDAS performs in the region. A formal validation is not necessary but at least some references describing its validity in the region are needed. See for instance, Spennemann et al., 2015 & 2020.**

We now reference studies that evaluate GLDAS performance in the area. Spennemann et al. (2015; 2020) reported that GLDAS reproduces the observed soil moisture patterns satisfactorily. They also found that land surface model derived indices using GLDAS can be used as soil monitoring indices in agricultural production management. Grings et al. (2015) evaluated satellite-derived estimations against in situ observations, suggesting that GLDAS seems to be a good soil moisture benchmark in the Pampas region.

The text in the revised manuscript states (lines 113 to 124):

" *In the absence of soil moisture observations, we employ products obtained from the Global Land Data Assimilation System (GLDAS; Rodell et al., 2004; Meng et al., 2012; Beaudoing and Rodell, 2019; 2020). GLDAS uses several land surface models to derive soil moisture from the surface water and energy balances forced by observations. The Noah Model is considered here. It has four soil layers (0-10 cm, 10-40 cm, 40-100 cm, and 100-200 cm) totaling 2 meters depth (Rodell et al., 2004). The total soil moisture in a column is the sum of the content in the four layers. The soil moisture data set consists of monthly values at a spatial resolution of 0.25°× 0.25° over the same period of analysis as precipitation. Evaluation of GLDAS soil moisture products in the Humid Pampas was recently performed by Grings et al. (2015) and Spennemann et al. (2015; 2020). According to Grings et al. (2015), GLDAS is a good soil moisture benchmark in the Pampas region since it achieved the highest correlation (r > 0.80) with in situ soil moisture measurements. Spennemann et al. (2015, 2020) also reported that GLDAS reproduces soil moisture observational patterns satisfactorily. They also found that GLDAS products can be used as soil monitoring indices in agricultural production management.*"

**16. Line 47: This only apply to dry events; wet events has a completely different dynamic. Please, clarify.**

We have removed the reference to wet periods from the sentence. It now states (lines 47 to 49): "*Statistical analyses of extreme events in SESA have shown that periods of water deficit can occur at different time scales, with an inverse relationship between frequency and duration, i.e., shorter-lived events tend to be more frequent than those of longer duration."*

**17. Line 69: Agriculture will be only one sector affected. Droughts can affect almost any compartment of economy and ecosystems.**

We have simplified the sentence to (lines 69-70): "*If confirmed, such change could be reflected in more droughts".*

**18. Line 104: the sentence between [: : :] should be removed of placed elsewhere.**

Thank you. The sentence has been removed.

**19. Line 203-205: how the soil moisture can be affected by soil degradation and desertification? Is this modelled in GLDAS?**

We have removed the speculative sentence and their references from the manuscript. GLDAS does not consider changes in soils and their properties, and therefore cannot address soil degradation issues. Recent studies have started looking into desertification with a combination of datasets that include GLDAS (e.g., Fan et al. 2020).

*20. Line 207: The SPI-6 could be correlated with hydrological droughts (probably longer aggregation periods can do it better) but it is still a meteorological indicator as relies only on precipitation. It is not accurate to attribute the representation of hydrological droughts to it. Please consider rephrasing this sentence.*

We agree with the reviewer. More extended periods like in SPI-6 can reflect better a hydrological drought than the shorter periods of SPI-3, but it indeed relies only on precipitation. The sentence removed the reference to hydrological drought (see lines 216 to 217).

"*Fig. 2d indicates that months with drought conditions, as represented by SPI6, are equivalent to 18% towards the region's north and southwest.*"

*21. Line 216-217: How the maps in Fig 3 can be interpreted as the temporal evolution of drought frequency?*

The sentence was removed. While the information was in the caption, the identification of the critical periods was not evident from the figure. We have added insets clarifying what part of the year each critical period corresponds (see Fig 3. at the end of the responses).

Wheat is sensitive to drought frequency in late spring (Oct-Nov), corn is sensitive to droughts during summer (Dec-Jan), and soybean is most sensitive during Jan-Feb. Therefore, it does not describe a temporal evolution of drought frequency, as our sentence claimed. It is the shift in relevant drought months for each crop type.

*22. Lines 314-315: "Therefore, droughts in the Core Crop Region are detected more easily when using SSI instead of SPI." It is not possible to benchmark drought detection by just comparing two drought indicators. Just because one indicator*

***covers more area or is more severe, etc., doesn't mean that is the best performing indicator. Instead, I would argue that the best indicator should be the one that better represents the specific sectorial impacts, in this case, crop production.***

Good point. We have rewritten the entire paragraph (see lines 302 to 311).

**References.**

*Beaudoing, H.,and Rodell, M. (2019). GLDAS Noah Land Surface Model L4 monthly 0.25 x 0.25 degree V2.0, Greenbelt, Maryland, USA, Goddard Earth Sciences Data and Information Services Center (GES DISC), NASA/GSFC/HSL, Accessed: [12 March 2019], https://doi.org/10.5067/9SQ1B3ZXP2C5.*

*Beaudoing, H., and Rodell, M. (2020). GLDAS Noah Land Surface Model L4 monthly 0.25 x 0.25 degree V2.1, Greenbelt, Maryland, USA, Goddard Earth Sciences Data and Information Services Center (GES DISC), NASA/GSFC/HSL, Accessed: [12 March 2019], https://doi.org/10.5067/SXAVCZFAQLNO.*

*Chen, M., Shi, W., Xie, P., Silva, V. B., Kousky, V. E., Wayne Higgins, R., & Janowiak, J. E. (2008). Assessing objective techniques for gauge-based analyses of global daily precipitation. Journal of Geophysical Research: Atmospheres, 113(D4).*

*Fan, Z.; Li, S. (2020). Fang, H. Explicitly Identifying the Desertification Change in CMREC Area Based on Multisource Remote Data. Remote Sens. 2020, 12, 3170. https://doi.org/10.3390/rs12193170.*

*Farahmand, A., and AghaKouchak, A. (2015). A generalized framework for deriving non-parametric standardized drought indicators. Adv. Water Resour, 76, 140-145, https://doi.org/10.1016/j.advwatres.2014.11.012.*

*Gringorten, I. I. (1963). A plotting rule for extreme probability paper. J. Geophys. Res., 68, 813-814, https://doi.org/10.1029/JZ068i003p00813.*

*Grings, F., Bruscantini, C. A., Smucler, E., Carballo, F., Dillon, M. E., Collini, E. A., Salvia, M., Karszenbaum, H. (2015). Validation strategies for satellite-based soil moisture products over Argentine Pampas. IEEE Journal of Selected Topics in Applied Earth Observations and Remote Sensing, 8(8), 4094-4105.*

*Holzman, M. E., Rivas, R., & Piccolo, M. C. (2014). Estimating soil moisture and the relationship with crop yield using surface temperature and vegetation index. International Journal of Applied Earth Observation and Geoinformation, 28, 181-192.*

*Llano, M.P. and Vargas, W. (2016), Climate characteristics and their relationship with soybean and maize yields in Argentina, Brazil and the United States. Int. J. Climatol., 36: 1471-1483. https://doi.org/10.1002/joc.4439.*

*Mallenahalli, N.K. (2020). Comparison of parametric and non-parametric standardized precipitation index for detecting meteorological drought over the Indian region. Theor Appl Climatol 142, 219–236. https://doi.org/10.1007/s00704-020-03296-z.*

*Massey Jr, F. J. (1951). The Kolmogorov-Smirnov test for goodness of fit. Journal of the American statistical Association, 46(253), 68-78.*

*Meng, J., Yang, R., Wei, H., Ek, M., Gayno, G., Xie, P., & Mitchell, K. (2012). The land surface analysis in the NCEP Climate Forecast System Reanalysis. Journal of Hydrometeorology, 13(5), 1621-1630.*

*Podestá, G., Bert, F., Rajagopalan, B., Apipattanavis, S., Laciana, C., Weber, E., ... & Menendez, A. (2009). Decadal climate variability in the Argentine Pampas: regional impacts of plausible climate scenarios on agricultural systems. Climate Research, 40(2-3), 199-210.*

*Rodell, M., Houser, P. R., Jambor, U. E. A., Gottschalck, J., Mitchell, K., Meng, C. J., ... & Entin, J. K. (2004). The global land data assimilation system. Bulletin of the American Meteorological Society, 85(3), 381-394.*

*Scian, B. V., & Bouza, M. E. (2005). Environmental variables related to wheat yields in the semiarid pampa region of Argentina. Journal of arid environments, 61(4), 669-679.*

*Seiler, R. A., Kogan, F., Wei, G., & Vinocur, M. (2007). Seasonal and interannual responses of the vegetation and production of crops in Cordoba–Argentina assessed by AVHRR derived vegetation indices. Advances in Space Research, 39(1), 88-94.*

*Silva, V. B., Kousky, V. E., & Higgins, R. W. (2011). Daily precipitation statistics for South America: An intercomparison between NCEP reanalyses and observations. Journal of Hydrometeorology, 12(1), 101-117.*

*Soláková T, De Michele C, Vezzoli R (2014). Comparison between parametric and non-parametric approaches for the calculation of two drought indices: SPI and SSI. J Hydrol Eng 19(9):04014010. https://doi.org/10.1061/(ASCE)HE.1943-5584.0000942.*

*Spennemann, P. C., Rivera, J. A., Saulo, A. C., & Penalba, O. C. (2015). A comparison of GLDAS soil moisture anomalies against standardized precipitation index and multisatellite estimations over South America. Journal of Hydrometeorology, 16(1), 158-171.*

*Spennemann, P. C., Fernández-Long, M. E., Gattinoni, N. N., Cammalleri, C., & Naumann, G. (2020). Soil moisture evaluation over the Argentine Pampas using models, satellite estimations and in-situ measurements. Journal of Hydrology: Regional Studies, 31, 100723.*

*Tallaksen, L. M., & Van Lanen, H. A. (Eds.). (2004). Hydrological drought: processes and estimation methods for streamflow and groundwater (Vol. 48). Elsevier.*

*Van Loon, A. F. (2015). Hydrological drought explained. Wiley Interdisciplinary Reviews: Water, 2(4), 359-392.* https://doi.org/10.1002/wat2.1085

*WMO, G.; GWP, G. Handbook of Drought Indicators and Indices. Geneva: World Meteorological Organization (WMO) and Global Water Partnership (GWP), 2016.*

|                        | SPI1 | SPI3 | SPI6 | SM Anomalies |
|------------------------|------|------|------|--------------|
| **Trend**              | -    | 3.2  | 5.5  | -            |
| **Quasi-cycle, T ~ 10 yr**  | 5.9  | 13.9 | 22   | 25.4         |
| **Quasi-cycle, T ~ 2.3 yr** | -    | 7.8  | 13   | 11.6         |

Table 1: Percentage of variance explained by the dominant modes of interannual variability detected using SSA with a window length of 120 months. Computations were done over SPI1, SPI3, SPI6, and soil moisture anomalies from January 1979 to December 2018 in Core Crop Region.

| Province | Crop | Indices | | | |
|----------|------|------|------|------|------|
| | | SPI3 | SPI6 | SSI3 | SSI6 |
| | wheat | 0.15 | 0.05 | 0.17 | 0.15 |
| Santa Fe | corn | 0.67 | 0.58 | 0.58 | 0.40 |
| | soybean | 0.68 | 0.58 | 0.62 | 0.52 |

| Province | Crop | Indices | | | |
|----------|------|------|------|------|------|
| | | SPI3 | SPI6 | SSI3 | SSI6 |
| | wheat | 0.49 | 0.04 | 0.44 | 0.38 |
| Córdoba | corn | 0.60 | 0.55 | 0.51 | 0.55 |
| | soybean | 0.73 | 0.70 | 0.58 | 0.70 |

**Table 3**: Correlation Coefficients of the annual detrended crop yield and the maximum or minimum index value for critical crop months (ON for wheat, DJ for corn, and JF for soybean). Maximum or minimum index values are identified according to whether detrended annual crop yields are negative or positive.

[Figure]

**Figure 2:** Percentage of months under moderate to extreme drought conditions (months below one standard deviation) of the total months from January 1979 to December 2018, according to (a) SPI1, (b) SPI3, (c) SPI6, and (d) soil moisture anomalies.

[Figure]

**Figure 3**: Percentage of months under moderate to extreme drought conditions (months below one standard deviation) during the crops' critical growing months from January 1979 to December 2018. For SPI3: (a) wheat during (Oct-Nov), (b) corn during (Dec-Jan), (c) soybean during (Jan-Feb). Panels (d), (e), and (f) are the same but for SPI6.

[Figure]

**Figure 4:** Areal-averaged time series from January 1979 to December 2018 for (a) SPI1, (b) SPI3, (c) SPI6, and (d) soil moisture anomalies in the Core Crop Region. The dominant modes of interannual variability are plotted in full lines.

[Figure]

**Figure 8:** The time series of the area-averaged annual crop yield over the provinces of Santa Fe and Córdoba from 1979 to 2018. (a) Wheat; (b) Corn; (c) Soybean. Cubic polynomial trends are shown in dotted lines. Panels d-f present the detrended yields for Santa Fe (blue) and Cordoba (orange). The superimposed gray bars characterize the SPI3 values corresponding to the crops' critical growing months: ON for wheat, DJ for corn, and JF for soybean.